# ContextNav: Towards Agentic Multimodal In-Context Learning

**Honghao Fu**[1], **Yuan Ouyang**[5], **Kai-Wei Chang**[4], **Yiwei Wang**[2], **Zi Huang**[1], **Yujun Cai**[1,3]*
[1]University of Queensland, [2]University of California, Merced, [3]Ant Group
[4]University of California, Los Angeles, [5]Nanjing University
honghao.fu@uq.edu.au, vanora.caiyj@gmail.com
https://contextnavpage.github.io/

## ABSTRACT

Recent advances demonstrate that multimodal large language models (MLLMs) exhibit strong multimodal in-context learning (ICL) capabilities, enabling them to adapt to novel vision-language tasks from a few contextual examples. However, existing ICL approaches face challenges in reconciling scalability with robustness across diverse tasks and noisy contextual examples: manually selecting examples produces clean contexts but is labor-intensive and task-specific, while similarity-based retrieval improves scalability but could introduce irrelevant or structurally inconsistent samples that degrade ICL performance. To address these limitations, we propose ContextNav, the first agentic framework that integrates the scalability of automated retrieval with the quality and adaptiveness of human-like curation, enabling noise-robust and dynamically optimized contextualization for multimodal ICL. ContextNav unifies context management and denoising within a closed-loop workflow driven by graph-based orchestration. Specifically, it builds a resource-aware multimodal embedding pipeline, maintains a retrievable vector database, and applies agentic retrieval and structural alignment to construct noise-resilient contexts. An Operational Grammar Graph (OGG) further supports adaptive workflow planning and optimization, enabling the agent to refine its operational strategies based on downstream ICL feedback. Experimental results demonstrate that ContextNav achieves state-of-the-art performance across various datasets, underscoring the promise of agentic workflows for advancing scalable and robust contextualization in multimodal ICL.

## 1 INTRODUCTION

In-context learning (ICL) has emerged as a fundamental capability of large language models, enabling adaptation to novel tasks through contextual demonstrations without parameter updates (Baldassini et al., 2024). By conditioning on task instructions and examples presented within the input context, ICL allows models to perform zero- or few-shot generalization without relying on gradient-based fine-tuning (Brown et al., 2020). This paradigm has been successfully extended to multimodal domains, where models leverage both textual and visual examples to generalize across vision-language tasks (Huang et al., 2025; Zhao et al., 2023; Li et al., 2024c; Doveh et al., 2024).

Existing multimodal ICL methods can be broadly divided into two categories: Manual ICL, where examples are manually selected and organized into contexts (Zhang et al., 2023; Sheng et al., 2024; Doveh et al., 2024), and Retrieval-based ICL, which employs feature embeddings to retrieve candidate examples as contexts (Gao et al., 2023; Suo et al., 2024; Chen et al., 2025a). While both approaches have shown promising results, they face notable challenges. Manual ICL often yields highly relevant and well-structured contexts but relies heavily on human curation, making it labor-intensive and difficult to generalize across large-scale multimodal corpora. Retrieval-based ICL alleviates this burden through automation; however, it may also retrieve semantically irrelevant samples and samples with inconsistent interrogative, imperative, or narrative structures, both of which can degrade

---

*Corresponding Author

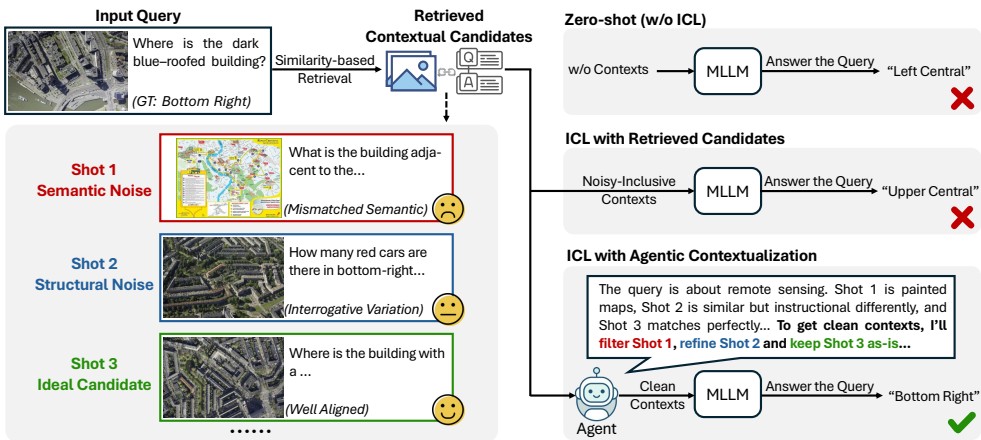

Figure 1: **Motivation for introducing agentic contextualization in multimodal ICL.** Similarity-based retrieval can introduce semantic or structural noise into contextual candidates, which degrades ICL effectiveness. Employing an agent for human-like curation could effectively alleviate this challenge. We provide a further discussion of the negative effects of such noise in Section 4.4, grounded in quantitative results.

downstream ICL performance. As illustrated in Figure 1, such noisy contexts can mislead models into producing incorrect answers.

Moreover, retrieval remains a static and rule-based one-shot process. Unlike human reasoning that adaptively refines example selection based on observed effectiveness, current systems cannot adapt their contextualization strategies or learn from experience. This creates a fundamental challenge: developing contextualization approaches that combine the scalability of automated retrieval with the quality and adaptiveness of human-like curation.

To address these challenges, we introduce ContextNav, an agentic framework for contextualization in multimodal ICL, as shown in Figure 2. Unlike previous methods, ContextNav formulates contextualization as an adaptive, tool-driven, and automated workflow that systematically manages multimodal context corpora, filters out noisy examples, reorganizes retrieved candidates in a human-like manner, and optimizes its tool orchestration strategies based on downstream ICL feedback. Specifically, ContextNav leverages the multimodal reasoning capability of the MLLM policy to build and maintain a resource-aware embedding pipeline with a retrievable vector database, and applies agentic retrieval and structural alignment to construct noise-resilient contexts. On top of these core components, an Operational Grammar Graph (OGG) provides a graph-structured action space that explicitly encodes the available operations along with their execution paths and dependency constraints. This structured action space allows the agent to plan and optimize its workflow adaptively: it can refine its operational strategy over multiple timesteps, continuously adjusting its choices based on downstream ICL feedback. Collectively, these components constitute a self-optimized closed-loop system that not only retrieves and manages contexts, but also adaptively organizes them, thereby enabling automated and robust multimodal ICL. ContextNav eliminates manual contextualization, achieving greater scalability than manual methods, while its human-like curation confers stronger noise robustness than retrieval-based approaches. To the best of our knowledge, ContextNav is the first agentic framework for contextualization in multimodal ICL. Our contributions are summarized as follows:

- We propose ContextNav, the first agentic framework that formulates multimodal contextualization as an adaptive, tool-driven, and fully automated agentic workflow, supporting scalable and denoisable contextualization for multimodal ICL.

- ContextNav transcends static one-shot retrieval by integrating the Operational Grammar Graph with a memory module that couples historical workflows and downstream ICL feedback, enabling adaptive optimization of workflow orchestration across timesteps and enhancing adaptability and robustness of contextualization.

- Through extensive experiments on diverse datasets, ContextNav achieves an average ICL gain of 16.8% across models, surpassing the prior state-of-the-art (7.6%) and underscoring the promise of agentic workflows for multimodal ICL.

## 2 RELATED WORKS

**In-Context Learning (ICL).** The concept of ICL was popularized by GPT-3 (Brown et al., 2020) and has since naturally emerged as a prominent paradigm in natural language processing (NLP) (Dong et al., 2022). A growing body of work has sought to understand the underlying mechanisms of ICL in LLM, primarily attributing this capability to implicit gradient descent (Dai et al., 2023a; Von Oswald et al., 2023; Chen et al., 2024) and Bayesian modeling frameworks (Xie et al., 2021; Arora et al., 2024; Wang et al., 2023; Falck et al., 2024). Concurrently, numerous studies have also focused on enhancing the ICL capabilities of LLMs by improving their inference frameworks (Chowdhery et al., 2023; Li et al., 2024a; Yang et al., 2024), training strategies (Sinha et al., 2024; Wu et al., 2025b), and contextualization methods (Yang et al., 2023; Wang et al., 2024a; Liu et al., 2023b), enabling more effective ICL on downstream tasks (Mao et al., 2025). These efforts have advanced both the theoretical understanding and practical application of ICL.

**Multimodal ICL.** The success of ICL in language models has spurred growing research interest in extending this paradigm to multimodal domains (Qin et al., 2024). Early MLLMs, such as Flamingo (Alayrac et al., 2022), InstructBLIP (Dai et al., 2023b), and LLaVA (Liu et al., 2023a), have demonstrated the potential of multimodal ICL. More recent studies further advance this capability by incorporating multimodal chain-of-thought reasoning (Zhang et al., 2023), vision expert models (Sheng et al., 2024), feature-based retrieval (Gao et al., 2023; Li et al., 2024b; Liu et al., 2023c; Tai et al., 2024), in-context tuning (Chen et al., 2023), representation engineering (Zhao et al., 2023; Li et al., 2025a; Huang et al., 2024), and attention editing (Li et al., 2025b). These techniques have collectively enhanced the zero- and few-shot ICL performance of VLMs across both specialized (Huang et al., 2025; Peng et al., 2024) and general vision-language tasks (Chen et al., 2025b). Despite these advances, current approaches still face challenges from contextual noise. In this paper, we address contextual noise by proposing an agentic workflow that combines the scalability of automated retrieval with the quality and adaptability of human-like curation, thereby enabling noise-robust and dynamically optimized contextualization.

## 3 CONTEXTNAV

### 3.1 OVERVIEW

We propose ContextNav, an agentic framework designed to advance the multimodal ICL performance of downstream MLLMs. Given a multimodal query as input, ContextNav establishes an end-to-end agentic pipeline that autonomously transforms raw corpora into well-formed, query-relevant contexts. As illustrated in Figure 2, the framework unfolds in three synergistic modules. First, the **Agentic Context Management** module constitutes the entry point. The agent performs resource-aware multimodal embedding, builds an evolving vector database, and retrieves a group of initial candidates from it given an input query. Second, the resulting candidates are feed into the **Context Denoising** module, where semantically and structurally noisy candidates are pruned or reorganized to yield cleaner contexts. In parallel, the **Graph-driven Workflow Orchestration** module oversees and coordinates these processes, ensuring that embedding, retrieval, and denoising operations form valid and optimized operation sequences. Collectively, these components establish an automated workflow for representing, managing, retrieving, and refining multimodal contexts, thereby supporting scalable and noise-resilient multimodal ICL for downstream MLLMs.

### 3.2 AGENTIC CONTEXT MANAGEMENT

**Resource-Aware Multimodal Embedding.** Embedding multimodal corpora is a prerequisite for building vector databases and enabling effective retrieval, while it faces several challenges. Large-scale embedding incurs heavy computational and storage costs, often becoming a system bottleneck. Embedding models also vary in accuracy and efficiency, creating trade-offs between fidelity and resource use, as shown in the experimental results of Table 3. Moreover, resource usage preference differs across users, requiring adaptive allocation. These factors motivate a resource-aware design that balances performance with efficiency in a demand-driven manner. To this end, ContextNav formulates multimodal embedding as an agent-driven process. The embedding-specification prompt $\mathcal{P}$emb (detailed in Appendix F and Appendix H) encodes the user's resource usage preferences, the

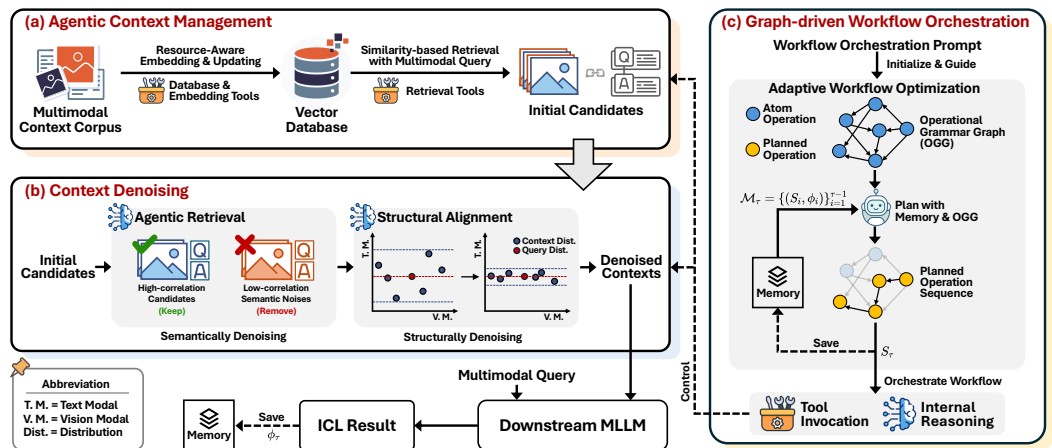

Figure 2: **Framework of the ContextNav.** The proposed agentic framework integrates three synergistic modules: (a) **Agentic Context Management**, which performs resource-aware multimodal embedding, maintains a continuously updated vector database, and subsequently leverages it for similarity-based retrieval to generate the initial candidate pool given an input query; (b) **Context Denoising**, which refines retrieved candidates through agentic retrieval and structural alignment to mitigate both semantic and structural noise; and (c) **Graph-driven Workflow Orchestration**, where the agent leverages an Operational Grammar Graph and memory module to adaptively plan and optimize operation sequence, thereby controlling the workflow. Collectively, these modules enable systematic management, representation, retrieval, and organization of multimodal contexts, supporting noise-robust and dynamically optimized contextualization for multimodal ICL.

current hardware state, and the embedding model zoo. Based on this prompt, the agent's MLLM policy $\pi_\theta$ employs its internal reasoning capability to perform resource-aware hardware-model matching, determining an appropriate pair of embedding models $(E_T, E_V) = \pi_\theta(\mathcal{P}_{\text{emb}})$, where $E_T$, $E_V$ denote textual and visual embedding models, respectively. Let the context corpus be denoted as $\mathcal{C} = \{(T_i, I_i)\}_{i=1}^N$, where $T_i$ is the $i$-th text instance and $I_i$ is its paired image. The agent leverages database and embedding tools to instantiate the embedding process at timestep $\tau$, which can be expressed as:

$$\mathcal{E}_\tau = \left\{ \left( E_{\text{text}}(T_i),\ E_{\text{vis}}(I_i) \right) \mid (T_i, I_i) \in \mathcal{C}_\tau \right\}, \tag{1}$$

where $\mathcal{E}_\tau$ denotes the multimodal embedding set constructed from the context corpus $\mathcal{C}_\tau$. In parallel, the agent employs database tools to continuously monitor the context corpus to identify newly added or modified samples $\Delta\mathcal{C}_{\tau+1}$ that have not been vectorized, thereby triggering an on-demand embedding pipeline for these samples in the subsequent timestep and yielding the corresponding embedding set $\Delta\mathcal{E}_{\tau+1}$. This ensures the database remains up-to-date.

**Vector Database Management.** Following the generation of embeddings, the agent undertakes the construction and adaptive management of a multimodal vector database $\mathcal{D} = \left\{ \left(T_i, I_i, e_i\right) \mid (T_i, I_i) \in \mathcal{C},\ e_i \in \mathcal{E} \right\}$. Contexts and their representations are systematically ingested, indexed, and archived, thereby establishing a structured and readily retrievable database. To accommodate corpus dynamics, the agent maintains an adaptive update mechanism with database tools:

$$\mathcal{D}_{\tau+1} = \mathcal{D}_\tau \cup \left\{ \left(T_j, I_j, \mathbf{e}_j\right) \mid (T_j, I_j) \in \Delta\mathcal{C}_{\tau+1},\ \mathbf{e}_j \in \Delta\mathcal{E}_{\tau+1} \right\}, \tag{2}$$

This agentic orchestration transforms the database from a static collection into an evolving knowledge structure that accurately reflects the state of the underlying context corpus.

**Initial Candidate Pool.** Once the vector database is constructed, the agent instantiates a Top-$k$ similarity-based retrieval function $f_\tau$, which is an adaptive composition of heterogeneous multimodal retrieval tools (textual, visual, or their cascaded combination). Given a budget $k$, the function yields the initial candidate pool:

$$R_\tau^{init}\ =\ f_\tau\big(q,\ \mathcal{D}_\tau,\ k\big), \tag{3}$$

where $q = (q_t, q_v)$ denotes a multimodal query composed of the textual query $q_t$ and its paired visual query $q_v$. The generated initial candidate pool is subsequently used for contextualization.

### 3.3 CONTEXT DENOISING

Within ContextNav, agentic retrieval and structural alignment are employed to attenuate semantic and structural noise from the initial candidates, respectively, with both processes grounded in the internal reasoning of the MLLM policy. Semantic noise refers to candidates whose content is off-topic or contradictory to the query intent, whereas structural noise arises when a candidate's interrogative, imperative or narrative structure diverges from that of the query.

**Agentic Retrieval.** As discussed in Section 1, initial candidates generated via similarity-based retrieval may contain both semantic and structural noise, which can undermine the downstream ICL performance. To mitigate the semantic noise introduced by weakly aligned matches, the agent subsequently applies a second-stage filtering beyond raw similarity search. Specifically, the coherence-specification prompt $P_{\text{coh}}$ (detailed in the Appendix F) encodes explicit instructions for semantic assessment, such as verifying topical consistency between query and candidate, and discarding candidates that contradict or distract from the query intent. Conditioned on this prompt, the MLLM policy $\pi_\theta$ leverages its internal reasoning capacity to evaluate each multimodal candidate in $R_\tau^{\text{init}}$ and decide whether to retain it. The process at timestep $\tau$ can be expressed as:

$$R_\tau^{sem} = \pi_\theta\big(q, \mathcal{P}_{\text{coh}}, R_\tau^{init}\big). \tag{4}$$

Through this agentic retrieval mechanism, the initial candidates are effectively denoised, ensuring the preservation of only those contexts that exhibit strong semantic alignment with the query.

**Structural Alignment.** Contextual candidates may exhibit variability in interrogative, imperative or narrative structure. Such heterogeneity introduces structural noise into the context, which prior work has extensively demonstrated to hinder consistent reasoning in ICL and lead to performance degradation (Voronov et al., 2024; Zhou et al., 2022; Zhao et al., 2021). To mitigate this, the agent applies a structure-refinement step that harmonizes the form of retrieved candidates with the query. Specifically, the structure-alignment prompt $\mathcal{P}_{\text{str}}$ (detailed in the Appendix F) encodes explicit instructions for reorganizing, ensuring that the textual flow mirrors that of the input query $q_t$. Conditioned on this prompt, the MLLM policy $\pi_\theta$ leverages its internal reasoning capability to edit candidates with divergent textual structures into a form consistent with $q_t$, thereby reducing structural discrepancies. The process at timestep $\tau$ can be expressed as:

$$R_\tau^{alin} = \pi_\theta\big(q_t, \mathcal{P}_{\text{str}}, R_\tau^{sem}\big). \tag{5}$$

This process aligns the textual structures of the candidates with that of the query, reducing distributional bias and structural noise. By combining semantic denoising and structural alignment, the agent yields a noise-minimized context set that enhances the robustness of contextualization.

### 3.4 GRAPH-DRIVEN WORKFLOW ORCHESTRATION

**Operational Grammar Graph (OGG).** In the proposed framework, the operations (tool invocation or internal reasoning) that the agent perform for context management and contextualization are governed by strict dependency relations and compositional constraints, since the underlying data structures evolve progressively within the workflow, with each stage reshaping the results of the previous one. Naive concatenation or heuristic composition risks redundant or invalid executions. Recent studies (Langchain-ai, 2024; Zhang et al., 2025a) demonstrate that graph structures effectively capture operational dependencies and control flows while enabling flexible multi-step workflows. Building on this insight, we construct a directed graph $\mathcal{G} = (\mathcal{V}, \mathcal{E})$, termed OGG, where $\mathcal{V}$ denotes the set of atomic operations and $\mathcal{E}$ encodes valid execution dependencies, thereby formalizing the grammar of permissible operations (detailed in the Appendix G). Specifically, the agent can instantiate a workflow that follows the operation sequence $S = (v_1 \rightarrow v_2 \rightarrow \cdots \rightarrow v_m)$, where each transition satisfies $(v_i, v_{i+1}) \in \mathcal{E}$, thus ensuring that the modules are executed in a valid order.

**Adaptive Workflow Optimization.** The agent plans and orchestrates workflows by exploiting the prior dependency structures encoded within the OGG $\mathcal{G}$. However, under the constraint of one-shot planning without intermediate feedback, the instantiated workflow may be suboptimal, potentially introducing inaccurate contextual candidates. This can reduce the number of valid contexts, shorten the effective context length, and ultimately degrade ICL performance, as illustrated in Figure 4. To address this, the agent adopts an adaptive optimization mechanism that leverages correlations between past workflow configurations and their downstream ICL performance stored in the memory

$\mathcal{M}$. Conditioned on a workflow orchestration prompt $\mathcal{P}_{\text{wop}}$ (detailed in Appendix F), workflow instantiation at timestep $\tau$ is modeled as the operation sequence:

$$S_\tau \;=\; \pi_\theta\big(\mathcal{P}_{\text{wop}}, \mathcal{M}_\tau, \mathcal{G}\big), \tag{6}$$

where $\mathcal{P}_{\text{wop}}$ specifies the workflow requirements at the initial timestep in the absence of downstream ICL feedback, and further declares the optimization logic governing subsequent timesteps. Notably, for operations that involve internal reasoning, the associated prompts (e.g., $\mathcal{P}_{\text{emb}}$, $\mathcal{P}_{\text{coh}}$, $\mathcal{P}_{\text{str}}$) are embedded internally within the operation itself rather than being directly specified in $\mathcal{P}$wop. This iterative optimization design enables the agent not only to leverage the OGG to enforce execution validity but also to adaptively refine orchestration strategies across multiple timesteps, thereby integrating context management and denoising into a coherent and adaptive pipeline.

### 3.5 Agentic Multimodal In-Context Learning

At each timestep $\tau$, the agent constructs a noise-minimized context set $R_\tau^{alin}$, which is concatenated with the multimodal query $q$ and fed into the downstream MLLM $\Phi$. The execution of in-context learning is guided by the prompt $\mathcal{P}_{\text{icl}}$ (detailed in the Appendix F). The model then produces both the final prediction $y_\tau$ for the input query and an auxiliary textual feedback $\phi_\tau$ that reflects the perceived quality of the constructed context. Formally, this process can be expressed as:

$$(y_\tau, \phi_\tau) \;=\; \Phi\big(R_\tau^{alin}, q, \mathcal{P}_{\text{icl}}\big), \tag{7}$$

The feedback $\phi_\tau$ provides an immediate assessment of context adequacy from the perspective of the MLLM performing ICL, thereby assisting the agent in optimizing its toolchain selection strategy. Specifically, the agent updates its memory $\mathcal{M}$ by recording the association between the executed operation sequence $S_\tau$ and the resulting feedback $\phi_\tau$:

$$\mathcal{M}_{\tau+1} \;=\; \mathcal{M}_\tau \cup \big\{(S_\tau, \phi_\tau)\big\} = \{(S_i, \phi_i)\}_{i=1}^\tau. \tag{8}$$

This continual feedback update closes the loop between multimodal ICL and adaptive toolchain optimization, enabling the agent to progressively refine its planning strategy to select toolchains that provide stronger contextual support and thereby enhance the robustness of multimodal ICL.

## 4 Experiment

### 4.1 Dataset and Implementation

**Dataset.** We first conduct a difficulty annotation of query samples from recent composite-task datasets and benchmarks, including BlindTest (Rahmanzadehgervi et al., 2024), MME-RealWorld (Zhang et al., 2025c), CharXiv (Wang et al., 2024c), GVL (Wei et al., 2024), and MathVision (Wang et al., 2024b). The annotation is determined by the accuracy of the models under evaluation: if more than half of the tested models answer a given query correctly, it is labeled as easy; otherwise, it is labeled as hard. Following a 3:7 sampling ratio (Clark et al., 2018) between easy and hard queries, we randomly sampled 803, 130, 100, 120, and 120 test instances from these datasets, respectively. The specificity of the BlindTest sampling size comes from its structure of seven sub-benches. To ensure that each sub-bench was sufficiently represented while still preserving the dataset's overall distribution, we sampled 10% from each. The remaining samples were used as support data for ICL. Additionally, we incorporated single-task datasets such as CLEVR (Johnson et al., 2017), FOMI (Vinyals et al., 2016), and TextOCR (Singh et al., 2021), adopting the test/support splits specified in VL-ICL Bench (Zong et al., 2025). Collectively, these datasets cover a broad spectrum of visual reasoning tasks—including abstract geometry, real-world scenes, charts, graphs, mathematics, spatial relations, counting, attributes and text recognition.

**Implementation.** We adopt Gemini-2.0-flash (GTeam et al., 2024) as the default MLLM policy for ContextNav. The open-source models involved in our experiments, including Phi-3.5V (Abdin et al., 2024), InternLMX2.5 (Cai et al., 2024), Qwen2.5-VL (Bai et al., 2025) and embedding models are deployed on an A100 80G GPU, while closed-source models, such as the Gemini series (Team et al., 2023) and GPT-4o (Hurst et al., 2024), are accessed via APIs on CPU servers. In addition, based on our experimental platform, ContextNav adopts Qwen3-Embedding-4B (Zhang et al., 2025b) as the textual embedding backbone and the vision encoder of Qwen2.5-VL (Bai et al., 2025) as the

Table 1: **Comparison of downstream accuracies with other baseline methods**, greater values indicate better performance. The bold numbers represent the best accuracy. 'Rand.' is the abbreviation for 'random'.

| Methods | BlindTest | RealWorld | CharXiv | GVL | MathVision | VL-ICL Bench | | | Average |
| --- | --- | --- | --- | --- | --- | --- | --- | --- | --- |
| | | | | | | CLEVR | FOMI | TextOCR | |
| Phi-3.5V-4.2B | 0.402 | 0.292 | 0.300 | 0.333 | **0.117** | 0.425 | 0.070 | 0.715 | 0.332 |
| +Rand. Sample | 0.339 | 0.254 | 0.250 | 0.308 | 0.092 | 0.395 | 0.070 | 0.685 | 0.299 |
| +VL-ICL | 0.407 | 0.292 | 0.280 | 0.325 | 0.100 | 0.435 | **0.080** | **0.745** | 0.333 |
| +MMICES | 0.399 | **0.300** | 0.300 | 0.317 | 0.083 | 0.415 | 0.070 | 0.730 | 0.327 |
| **+ContextNav** | **0.443** | **0.300** | **0.310** | **0.350** | **0.117** | **0.440** | 0.070 | 0.740 | **0.346** |
| InternLMX2.5-7B | 0.303 | 0.262 | 0.200 | 0.325 | 0.108 | 0.545 | 0.075 | 0.475 | 0.287 |
| +Rand. Sample | 0.288 | 0.231 | 0.150 | 0.292 | 0.083 | 0.505 | 0.075 | 0.445 | 0.259 |
| +VL-ICL | 0.316 | 0.246 | 0.180 | **0.342** | 0.092 | **0.570** | **0.105** | 0.455 | 0.288 |
| +MMICES | 0.296 | 0.254 | 0.210 | 0.333 | 0.092 | 0.555 | 0.090 | 0.465 | 0.287 |
| **+ContextNav** | **0.358** | **0.277** | **0.230** | **0.342** | **0.142** | **0.570** | 0.100 | **0.480** | **0.312** |
| Qwen2.5-VL-7B | 0.566 | 0.307 | 0.390 | 0.342 | 0.217 | 0.820 | 0.045 | 0.835 | 0.440 |
| +Rand. Sample | 0.496 | 0.277 | 0.340 | 0.300 | 0.200 | 0.785 | 0.045 | 0.820 | 0.408 |
| +VL-ICL | 0.594 | 0.323 | 0.370 | 0.358 | 0.233 | 0.900 | **0.060** | **0.845** | 0.460 |
| +MMICES | 0.606 | 0.315 | 0.390 | 0.342 | 0.208 | 0.890 | 0.050 | **0.845** | 0.456 |
| **+ContextNav** | **0.645** | **0.338** | **0.400** | **0.367** | **0.250** | **0.940** | 0.055 | **0.845** | **0.480** |
| Gemini-1.5-flash | 0.755 | 0.300 | 0.490 | 0.475 | 0.483 | 0.535 | 0.120 | 0.880 | 0.505 |
| +Rand. Sample | 0.733 | 0.285 | 0.440 | 0.492 | 0.350 | 0.510 | 0.140 | 0.860 | 0.476 |
| +VL-ICL | 0.775 | 0.315 | 0.500 | 0.533 | 0.475 | 0.555 | 0.180 | **0.890** | 0.528 |
| +MMICES | 0.802 | 0.338 | 0.520 | 0.508 | 0.467 | 0.540 | 0.170 | 0.875 | 0.528 |
| **+ContextNav** | **0.859** | **0.369** | **0.570** | **0.575** | **0.517** | **0.575** | **0.240** | **0.890** | **0.574** |
| Gemini-2.0-flash | 0.761 | 0.308 | 0.510 | 0.508 | 0.492 | 0.775 | 0.080 | 0.900 | 0.542 |
| +Rand. Sample | 0.733 | 0.292 | 0.430 | 0.517 | 0.341 | 0.755 | 0.095 | 0.870 | 0.504 |
| +VL-ICL | 0.773 | 0.331 | 0.490 | 0.550 | 0.483 | 0.810 | 0.145 | 0.905 | 0.561 |
| +MMICES | 0.800 | 0.338 | 0.520 | 0.525 | 0.467 | 0.795 | 0.140 | 0.885 | 0.559 |
| **+ContextNav** | **0.854** | **0.377** | **0.560** | **0.600** | **0.550** | **0.825** | **0.155** | **0.910** | **0.604** |
| GPT-4o | 0.609 | 0.323 | 0.530 | 0.500 | 0.342 | 0.610 | 0.085 | 0.870 | 0.484 |
| +Rand. Sample | 0.588 | 0.308 | 0.470 | 0.517 | 0.308 | 0.635 | 0.095 | 0.860 | 0.473 |
| +VL-ICL | 0.631 | 0.338 | 0.540 | 0.558 | 0.333 | 0.650 | 0.140 | 0.890 | 0.510 |
| +MMICES | 0.649 | 0.353 | 0.550 | 0.542 | 0.316 | 0.655 | 0.150 | 0.895 | 0.514 |
| **+ContextNav** | **0.672** | **0.392** | **0.580** | **0.608** | **0.383** | **0.670** | **0.165** | **0.905** | **0.547** |

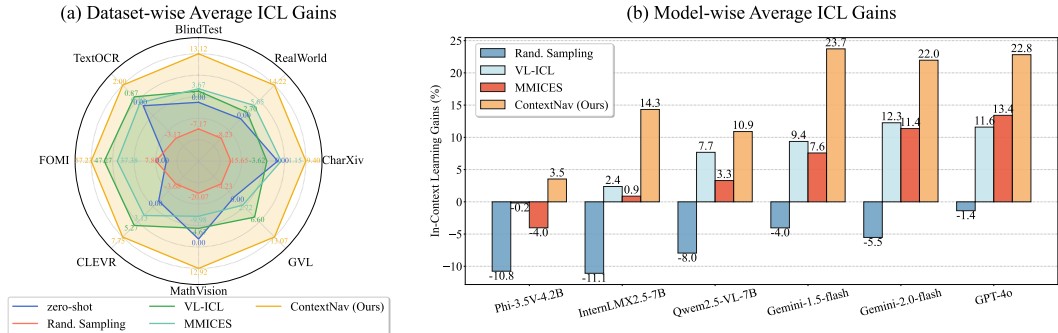

Figure 3: Comparison of average ICL gains with baselines. (a) Dataset-wise average gains across 8 datasets. (b) Model-wise average gains across 6 representative MLLMs. 'ICL Gains' represents the percentage improvement of an MLLM (zero-shot) after applying ICL. The complete comparative results are provided in Appendix B.

visual embedding backbone. Based on the experimental results of Figure 4, the framework employs 8 effective shots for ICL. Unless otherwise specified, experimental results are the medium number of 5 trials, and ablation experiments are conducted on the MathVision. The prompts involved in the agentic system are documented in Appendix F, while the definitions of the tool library and the OGG are provided in Appendix G, and the default embedding model zoos are listed in Appendix H. In addition, Appendix I presents case studies of both successful and failure examples.

## 4.2 COMPARISON AGAINST OTHER METHODS

We compare ContextNav with zero-shot and multimodal ICL baselines under training-free settings, including random sampling, VL-ICL (Zong et al., 2025), and MMICES (Chen et al., 2025b). The

Table 2: **Ablation study on agent policy and core components involved in contextualization.** The bold entries indicate the best results. 'Semantic Noise' and 'Structural Noise' denote the proportions of contextual candidates with the corresponding type of noise, respectively. 'ICL Gains' represents the percentage improvement of a vanilla MLLM after applying ICL. 'Rand.' is the abbreviation of random. 'X-Step TSR' represents the probability of successfully generating a valid toolchain within 'X' iterations. '–' indicates not applicable. As Agentic Retrieval enforces semantic alignment, it is disabled when ablating textual/visual retrieval to avoid diminishing the significance of the results.

| Methods | Semantic Noise ↓ | Structural Noise ↓ | ICL Gains ↑ | 1-Step TSR ↑ | 5-Step TSR ↑ |
|---|---|---|---|---|---|
| *MLLM policy* | | | | | |
| GPT-4o | **0.053** | **0.081** | **+11.8%** | **1.000** | **1.000** |
| Gemini-2.0-flash | **0.053** | 0.084 | **+11.8%** | 0.995 | **1.000** |
| Qwen2.5-VL-7B | 0.073 | 0.107 | +10.1% | 0.985 | **1.000** |
| Qwen2.5-VL-3B | 0.080 | 0.139 | +8.4% | 0.965 | 0.995 |
| *Other Components* | | | | | |
| w/o Agentic Retrieval (AR) | 0.171 | 0.090 | +1.6% | 0.995 | **1.000** |
| w/o Structural Alignment | **0.053** | 0.573 | +6.7% | 0.995 | **1.000** |
| w/o Textual Retrieval Tools & AR | 0.433 | 0.143 | -18.7% | 0.995 | **1.000** |
| w/o Visual Retrieval Tools & AR | 0.249 | **0.076** | -3.5% | 0.995 | **1.000** |
| w/o Toolchain Optimization | 0.093 | 0.091 | +5.0% | 0.995 | **1.000** |
| w/o Operational Grammar Graph | - | - | - | 0 | 0 |
| **Full (with Gemini-2.0-flash)** | **0.053** | 0.084 | **+11.8%** | 0.995 | **1.000** |

Table 3: **Ablation study of embedding models.** 'Semantic Noise' and 'Structural Noise' w/o AR & SA denote the proportions of contextual candidates with the corresponding type of noise in the absence of Agentic Retrieval (AR) and Structural Alignment (SA). 'Effective Rate' refers to the proportion of retrieved candidates retained after applying Agentic Retrieval. 'ICL Gains' represents the percentage improvement of a vanilla MLLM after ICL.

| Text Encoder | Vision Encoder | Semantic Noise ↓ w/o AR & SA | Structural Noise ↓ w/o AR & SA | Effective Rate ↑ | ICL Gains ↑ |
|---|---|---|---|---|---|
| **Qwen3-Embedding-8B** | **Qwen2.5-VL-VisEnc** | **0.168** | **0.581** | **0.765** | **+11.8%** |
| | CLIP-vis | 0.215 | 0.610 | 0.719 | +10.1% |
| Qwen3-Embedding-4B | Qwen2.5-VL-VisEnc | 0.171 | 0.584 | 0.762 | **+11.8%** |
| | CLIP-vis | 0.216 | 0.611 | 0.718 | +10.1% |
| Qwen3-Embedding-0.6B | Qwen2.5-VL-VisEnc | 0.194 | 0.595 | 0.749 | **+11.8%** |
| | CLIP-vis | 0.227 | 0.623 | 0.721 | +10.1% |
| CLIP-text | Qwen2.5-VL-VisEnc | 0.276 | 0.631 | 0.652 | +8.4% |
| | CLIP-vis | 0.311 | 0.659 | 0.606 | +6.7% |

results of VL-ICL and MMICES are obtained from our replications based on their original methodologies. Specifically, the replication of VL-ICL follows the procedure of constructing a manually coarse-filtered candidate pool, followed by random sampling, whereas MMICES adopts a cascaded retrieval process that prioritizes visual retrieval and subsequently applies text-based retrieval. The results in Table 1 and Figure 3 demonstrate that ContextNav rivals or surpasses other carefully designed baselines across nearly all datasets and MLLMs. When the support data involves composite tasks and becomes more complex and noisy (e.g., BlindTest, RealWorld, CharXiv, and MathVision), non-agentic methods are more likely to yield unstable or even degraded performance, whereas ContextNav consistently improves MLLMs' performance. Overall, ContextNav achieves an average ICL gain of 16.8% across models and 16.2% across datasets, substantially outperforming the previous state-of-the-art (7.6% and 8.2%, respectively). These results further highlight that ContextNav provides a noise-robust mechanism for exploiting in-context information, delivering consistent and substantial ICL gains across diverse multimodal tasks. Moreover, we extend the comparison results in Appendix C by incorporating additional baselines, including many-shot ICL (Agarwal et al., 2024) and Cache of Thought (Wu et al., 2025a), following the settings used in their original works.

## 4.3 ABLATION STUDY

**Agent policy and core components involved in contextualization.** The upper part of Table 2 indicates that the choice of MLLM policy could affect the effectiveness of ContextNav. More capable

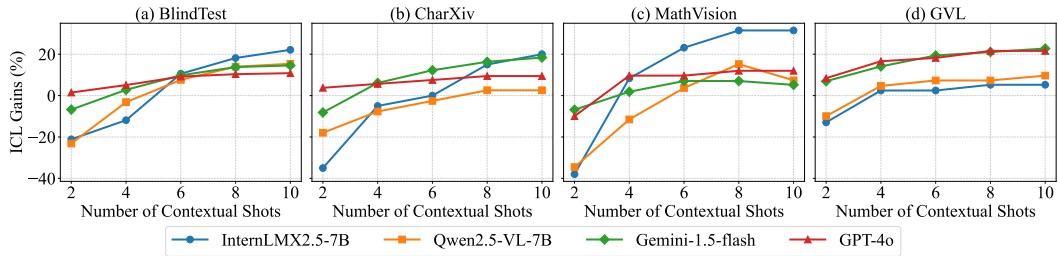

Figure 4: **Ablation study on the number of contextual shots.** 'ICL Gains' represents the percentage improvement after ICL.

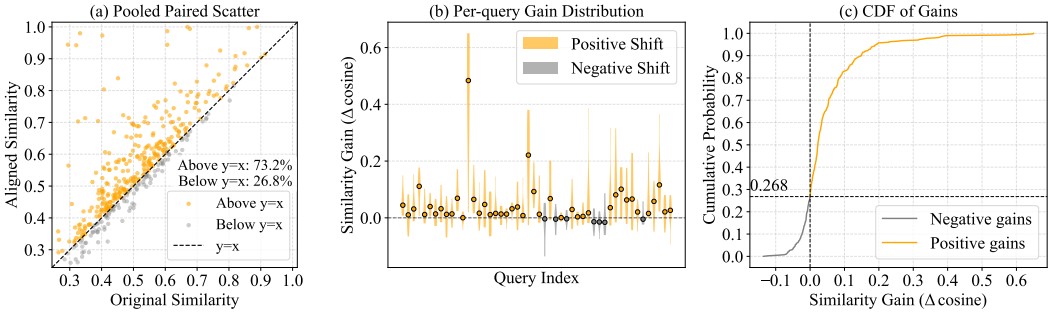

Figure 5: **Effect of Structural Alignment on textual similarity distributions.** (a) Paired scatter between original and aligned similarities. (b) Per-query distributions of similarity gains, with orange denoting positive shifts and gray denoting non-positive shifts, and the points indicate the average gains. (c) Cumulative Distribution Function (CDF) of similarity gains, with reference lines indicating zero-gain boundaries.

multimodal policies demonstrate stronger capacity for executing agentic retrieval and structural alignment, thereby reducing both semantic and structural noise and yielding higher ICL gains. In addition, stronger models exhibit improved instruction-following ability, which translates into more reliable toolchain generation and execution. The lower part of the table highlights the contribution of individual modules. The ablation results highlight that textual, visual, and agentic retrieval modules are crucial, as their removal markedly amplifies semantic noise. Semantic noise emerges as the dominant factor shaping ICL gains, exerting a stronger impact than other sources of disturbance. At the same time, structural alignment plays a key role in mitigating structural noise, whose influence on ICL, though secondary to semantic noise, remains non-negligible. Moreover, disabling toolchain optimization and restricting the workflow to a single-step determination could yield suboptimal toolchain selection strategies, which consequently limit the effective mitigation of both semantic and structural noise and thereby reduce the ICL gains. In addition, the OGG constitutes the foundation of tool orchestration, and its removal renders the system incapable of executing valid toolchains.

**Embedding.** As shown in Table 3, we conduct a manual ablation of the embedding models in ContextNav. For the text encoder, we evaluate the Qwen3-Embedding series at different parameters, the SigLIP-2 (Tschannen et al., 2025), and the classical CLIP (Radford et al., 2021). For the vision encoder, we test the language-aligned Qwen2.5-VL visual transformer, SigLIP-2 and CLIP. We observe that embeddings from models with fewer parameters lead to less accurate retrieval, introducing noisy or suboptimal candidates. While noisy candidates are filtered out in Agentic Retrieval, suboptimal ones may remain and still degrade contextual quality, diminishing ICL gains and highlighting the importance of careful embedding selection. We also find that Qwen3-Embedding-8B and 4B yield nearly identical retrieval under the default setting, showing that indiscriminately adopting larger models may bring diminishing returns and unnecessary resource overhead. This finding highlights the practical significance of ContextNav's resource-aware design, which adaptively selects embedding models according to both the user's resource usage preferences and the objective hardware constraints.

**Number of contextual shots.** As shown in Figure 4, we conduct an ablation study on four MLLMs across four datasets to examine the impact of contextual shot numbers on ICL gains. The results

indicate that ICL gains generally improve with more shots but plateaus as the number increases. Smaller models (e.g., Qwen2.5VL-7B and InternLMX2.5-7B) are more sensitive, with too few shots could yield negative gains, whereas larger models (e.g., Gemini-1.5-flash and GPT-4o) exhibit more stable improvements. These results highlight the need for an appropriate choice of shot number: too few degrade performance, while too many add computational cost without commensurate benefit. Accordingly, we set the default to 8 shots in our experiments.

**Extended ablations.** In Appendix D, we extend our analysis by incorporating additional ablations on noise robustness (D.1), the semantic equivalence of structural alignment (D.2), the interaction between MLLM policies and system cost (D.3), as well as the trade-off between ICL gains and system costs for downstream MLLMs (D.4). These additional studies further demonstrate the overall robustness, effectiveness, and scalability of ContextNav across diverse settings.

### 4.4 DISCUSSION

**Structural Alignment.** As shown in Figure 5, we analyze the effect of Structural Alignment on textual similarity distributions. We randomly sample 50 queries along with their eight most similar candidates from MATH-Vision dataset and apply structural alignment to them. Panel (a) compares original and refined textual similarities, showing that 73.2% of the points lie above the diagonal, indicating that aligned generally increases similarity. Panel (b) illustrates per-query distributions of similarity gains, where most queries exhibit a positive shift; although a few negative gains remain, they are marginal in magnitude. Panel (c) presents the cumulative distribution of similarity gains, further confirming that a substantial proportion of candidates benefit from refinement. Overall, these results demonstrate that Structural Alignment mitigates structural discrepancies between queries and candidates, leading to more consistent and semantically aligned contexts.

**Negative effects of noisy contexts.** As shown in Table 1, contextualization with random sampling strategies often results in substantial performance degradation. This decline arises because random sampling introduces a large number of query-irrelevant examples, thereby injecting noise into the context. Furthermore, as reported in Table 2, higher proportions of semantic and structural noise generally correspond to reduced ICL gains. These findings underscore the detrimental impact of noisy contexts in ICL and highlight the necessity of accurate exemplar retrieval. This observation is also consistent with the implicit gradient descent perspective of ICL (Dai et al., 2023a; Von Oswald et al., 2023; Chen et al., 2024), in which irrelevant or misaligned examples distort the optimization trajectory and hinder generalization.

**Extended discussion.** We further elaborate our discussion in Appendix E on the dataset selection( E.1), construction process and robustness of the Operational Grammar Graph (E.2), provide an analysis of system-level failure cases (E.3), and discuss potential future research directions for agentic multimodal ICL (E.4).

**Limitations.** Since the agent requires additional inference and tool-execution steps, ContextNav inevitably introduces extra token overhead and system latency, which may limit its applicability in scenarios with stringent real-time requirements. On average, each ICL iteration under the default setting consumes 22.51K tokens and incurs 3.26 seconds of delay. As a complement, Appendix D.3 reports the token overhead and system latency when using different MLLMs.

## 5 CONCLUSION

In this paper, we introduce ContextNav, the first agentic framework that integrates automated retrieval with human-like curation for multimodal ICL. By combining agentic context management, context denoising, and graph-based workflow orchestration modules, ContextNav constructs and optimizes noise-resilient contexts within a fully automated workflow, enhancing the multimodal ICL performance of downstream MLLMs. Experiments demonstrate that ContextNav achieves state-of-the-art results across diverse datasets and models, underscoring the potential of agentic workflows for scalable, adaptive, and robust contextualization in multimodal ICL.

## ACKNOWLEDGMENTS

The work is partially supported by the U.S. National Science Foundation (NSF) Grant CRII 2451683, an NVIDIA Academic Grants Program, a U.S. Bank Academic Research Award, the University of California, Merced, and a UC Merced Faculty Research Award. The views and conclusions are those of the authors and do not necessarily reflect the official policy or position of the U.S. Government.

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

## A  THE USE OF LARGE LANGUAGE MODELS (LLMS)

The paper employs LLMs for language polishing.

## B  EXTENTED MAIN RESULTS

Table 4 presents the complete ICL gain comparison results.

Table 4: **Comparison of ICL gains (%) with other baseline methods.** ICL gain represents the percentage improvement of a vanilla MLLM after ICL, greater values indicate better performance.

| Downstream MLLM | Methods | BlindTest | RealWorld | CharXiv | GVL | MathVision | CLEVR | FOMI | TextOCR | Mean |
|---|---|---|---|---|---|---|---|---|---|---|
| Phi-3.5V-4.2B | Rand. Sample | -15.6 | -13.0 | -16.7 | -8.1 | -21.4 | -7.1 | 0 | -4.2 | -10.8 |
| | VL-ICL | 1.2 | 0 | -6.6 | -2.4 | -14.5 | 2.4 | **14.3** | 4.2 | -0.2 |
| | MMICES | -0.7 | **2.7** | 0 | -4.8 | -29.1 | -2.4 | 0 | 2.1 | -4.0 |
| | **ContextNav** | **10.2** | **2.7** | **3.3** | **5.1** | 0 | **3.5** | 0 | **3.5** | **3.5** |
| InternLMX2.5-7B | Rand. Sample | -5.0 | -11.8 | -25.0 | -10.2 | -23.1 | -7.3 | 0 | -6.3 | -11.1 |
| | VL-ICL | 4.3 | -6.1 | -10.0 | **5.2** | -14.8 | **4.6** | **40.0** | -4.2 | 2.4 |
| | MMICES | -2.3 | -3.1 | 5.0 | 2.5 | -14.8 | 1.8 | 20.0 | -2.1 | 0.9 |
| | **ContextNav** | **18.2** | **5.7** | **15.0** | **5.2** | **31.5** | **4.6** | 33.3 | **1.1** | **14.3** |
| Qwen2.5-VL-7B | Rand. Sample | -12.4 | -9.8 | -15.0 | -12.3 | -7.8 | -4.5 | 0 | -1.8 | -8.0 |
| | VL-ICL | 4.9 | 5.2 | -5.1 | 4.7 | 7.4 | 9.8 | **33.3** | **1.2** | 7.7 |
| | MMICES | 7.1 | 2.6 | 0 | 0 | -4.1 | 8.5 | 11.1 | **1.2** | 3.3 |
| | **ContextNav** | **14.0** | **10.1** | **2.6** | **7.3** | **15.2** | **14.6** | 22.2 | **1.2** | **10.9** |
| Gemini-1.5-flash | Rand. Sample | -2.9 | -5.0 | -10.2 | 3.6 | -27.5 | -4.7 | 16.7 | -2.3 | -4.0 |
| | VL-ICL | 2.6 | 5.0 | 2.0 | 12.2 | -1.7 | 3.7 | 50.0 | **1.1** | 9.4 |
| | MMICES | 6.2 | 12.7 | -3.9 | 6.9 | -3.3 | 0.9 | 41.7 | -0.6 | 11.4 |
| | **ContextNav** | **13.8** | **23.0** | **16.3** | **21.1** | **7.0** | **7.5** | **100.0** | **1.1** | **23.7** |
| Gemini-2.0-flash | Rand. Sample | -3.7 | -5.2 | -15.7 | -1.8 | -30.7 | -2.6 | 18.8 | -3.3 | -5.5 |
| | VL-ICL | 1.6 | 7.5 | -3.9 | 8.3 | -1.8 | 4.5 | 81.3 | 0.6 | 16.4 |
| | MMICES | 5.1 | 9.7 | 2.0 | 3.3 | -5.1 | 2.6 | 75.0 | -1.7 | 11.4 |
| | **ContextNav** | **12.2** | **22.4** | **9.8** | **18.1** | **11.8** | **6.5** | **93.8** | **1.1** | **22.0** |
| GPT-4o | Rand. Sample | -3.4 | -4.6 | -11.3 | 3.4 | -9.9 | 4.1 | 11.8 | -1.1 | -1.4 |
| | VL-ICL | 3.6 | 4.6 | 1.9 | 11.6 | -2.6 | 6.6 | 64.7 | 2.3 | 11.6 |
| | MMICES | 6.6 | 9.3 | 3.8 | 8.4 | -7.6 | 7.4 | 76.5 | 2.9 | 13.4 |
| | **ContextNav** | **10.3** | **21.4** | **9.4** | **21.6** | **12.0** | **9.8** | **94.1** | **4.0** | **22.8** |

Table 5: Comparison with many-shot ICL.

| MLLM | Method (+sample num) | CLEVR ↑ | MathVision ↑ | CharXiv ↑ |
|---|---|---|---|---|
| **InternLMX2.5-7B** | zero-shot | 0.545 | 0.108 | 0.200 |
| | Random (+8) | 0.505 | 0.083 | 0.150 |
| | Many-shot (+100) | 0.535 | 0.092 | 0.160 |
| | **ContextNav (+8)** | **0.570** | **0.142** | **0.230** |
| **Qwen2.5-VL-7B** | zero-shot | 0.820 | 0.217 | 0.390 |
| | Random (+8) | 0.785 | 0.200 | 0.340 |
| | Many-shot (+100) | 0.870 | 0.208 | 0.370 |
| | **ContextNav (+8)** | **0.940** | **0.250** | **0.400** |
| **GPT-4o** | zero-shot | 0.610 | 0.342 | 0.530 |
| | Random (+8) | 0.635 | 0.308 | 0.470 |
| | Many-shot (+100) | 0.645 | 0.316 | 0.510 |
| | **ContextNav (+8)** | **0.670** | **0.383** | **0.580** |
| **Gemini-2.0-flash** | zero-shot | 0.775 | 0.492 | 0.510 |
| | Random (+8) | 0.755 | 0.341 | 0.430 |
| | Many-shot (+800) | 0.815 | 0.425 | 0.500 |
| | **ContextNav (+8)** | **0.825** | **0.550** | **0.560** |

## C  EXTENDED COMPARISON

### C.1  COMPARISON IN MANY-SHOT CONDITIONS

We further extend the quantitative comparison with many-shot ICL (Agarwal et al., 2024)) in Table 5. Since different models have different context-length limits, we use as many in-context samples as allowed by either the model's maximum window or the corpus upper bound (e.g., 800 samples for CLEVR in VL-ICL). For models whose context window becomes the bottleneck (e.g., InternLM, Qwen2.5-VL, GPT-4o), we follow the standard many-shot ICL protocol and randomly sample instances from the corpus.

From our experiments, we find that increasing the number of randomly sampled examples from 8 to 100 or even 800 generally yields a positive shift in downstream MLLM ICL performance. On the clean, single-task benchmark CLEVR, most MLLMs move from negative to positive gains as the support set grows. However, on more complex and noisier benchmarks such as MathVision and CharXiv, many-shot random sampling improves over few-shot (8) but still remains below zero-shot performance. Across all settings, ContextNav consistently performs best. These findings show that both enlarging the support set and selecting query-relevant examples have positive effects on ICL.

### C.2  COMPARISON WITH CACHE OF THOUGHT

We extended a comparison with Cache of Thought (Wu et al., 2025a) in Table 6. We compare ContextNav with Cache of Thought using the settings reported in its original paper, evaluating both methods with OpenFlamingo-3B on CLEVR and TextOCR. Under the strict 1–2 shot setting, ContextNav performs slightly worse. This is expected: Cache of Thought distills both knowledge and explicit reasoning traces from a powerful Master VLM, whereas ContextNav retrieves relevant evidence from a corpus without providing explicit chains of thought. As a result, Cache of Thought has an inherent advantage when only one or two demonstrations are available.

## D  EXTENDED ABLATIONS

### D.1  PERFORMANCE ON NOISY CORPUS

To further quantify the noise level of the corpora and evaluate ContextNav's robustness to noisy data, we conducted additional experiments on MathVision and on a large-scale real-world noisy corpus, Conceptual Captions (CC12M) (Changpinyo et al., 2021). For CC12M, we sampled 80K image–caption pairs and automatically rewrote them into a question–answer format using GPT-5. A

Table 6: Comparison with Cache of Thought on OpenFlamingo-3B.

| Method | CLEVR ↑ | TextOCR ↑ |
|---|---|---|
| **zero-shot** | 7.50 | 0.00 |
| **1-shot** | | |
| Cache of Thought | 20.50 | 6.50 |
| ContextNav | 12.00 | 3.50 |
| **2-shot** | | |
| Cache of Thought | 28.00 | 5.50 |
| ContextNav | 15.50 | 4.50 |

Table 7: Semantic and structural noise under different ablation settings on MathVision and CC12M.

| Method | MathVision | | CC12M | |
|---|---|---|---|---|
| | Semantic Noise ↓ | Structural Noise ↓ | Semantic Noise ↓ | Structural Noise ↓ |
| w/o Agentic Retrieval (AR) | 0.171 | 0.090 | 0.159 | 0.077 |
| w/o Structural Alignment (SA) | 0.053 | 0.573 | 0.050 | 0.488 |
| w/o Textual Retrieval Tools & AR | 0.433 | 0.143 | 0.552 | 0.161 |
| w/o Visual Retrieval Tools & AR | 0.249 | 0.076 | 0.206 | 0.072 |
| Rand. Sampling | 0.936 | 0.971 | 0.987 | 0.960 |
| **Full (ContextNav)** | **0.053** | **0.084** | **0.050** | **0.075** |

200-sample evaluation set was then curated following the same selection criteria used in our other benchmarks, and all evaluation samples were verified to have sufficiently many support examples within the corpus. As shown in Table 7, random sampling yields highly noisy retrieved contexts, confirming the substantial intrinsic noise present in both MathVision and CC12M. The results further demonstrate the reliability of each component in our method for mitigating such noise, as well as the overall noise robustness of our approach.

## D.2 SEMANTIC EQUIVALENCE OF STRUCTURAL ALIGNMENT

To verify the reliability of structural alignment, we conduct an empirical evaluation of semantic equivalence on the MathVision. We quantify semantic equivalence using two complementary metrics: (1) the semantic similarity between the original query and its aligned counterpart, computed using Qwen3-Embedding-4B; and (2) a binary classification accuracy from GPT-4o, which assesses whether the aligned structure preserves the original meaning. As shown in Table 8, structural alignment preserves a high level of semantic equivalence and, for Qwen2.5-VL-7B and stronger policy LLMs, rarely leads to any semantic distortion (GPT-4o accuracy = 1). The prompt used for GPT-4o accuracy evaluation is:

---
**Prompt for GPT-4o Accuracy Evaluation**

You will be given two texts, and please determine whether they convey the same meaning, ignoring superficial differences in wording, structure, or format. If the meanings are semantically equivalent, answer "Yes". If the meanings differ in any essential way, answer "No". The texts are below:

Text 1 {query}

Text 2 {aligned_query}

Answer with "Yes" or "No" only.

---

## D.3 MLLM POLICIES AND SYSTEM COSTS

Table 9 provides a comprehensive comparison of different MLLM policies within ContextNav, including their token overhead, system delay, noise reduction capability, and resulting ICL gains. Motivated by a balanced consideration of performance, latency, and cost, we use Gemini-2.0-flash as our standard implementation. Empirically, Gemini-2.0-flash delivers strong performance while achieving substantially lower latency than alternative models. As an API-based solution, it also avoids

Table 8: Evaluation of semantic equivalence before and after structural alignment.

| Policy LMM | Semantic Similarity ↑ | GPT-4o Accuracy ↑ |
|---|---|---|
| Qwen2.5-VL-3B | 0.8946 | 0.983 |
| Qwen2.5-VL-7B | 0.9247 | 1.000 |
| Gemini-2.0-flash | 0.9313 | 1.000 |

Table 9: Comparison of policy models on noise reduction, ICL gains, and system efficiency.

| Policy | Semantic Noise ↓ | Structural Noise ↓ | ICL Gains ↑ | System Latency (s) ↓ | Token Costs ↓ | API Pricing ↓ |
|---|---|---|---|---|---|---|
| Qwen2.5-VL-3B | 0.080 | 0.139 | +8.4% | 5.10 | 22.2K | – |
| Qwen2.5-VL-7B | 0.073 | 0.107 | +10.1% | 7.38 | 22.6K | – |
| Gemini-2.0-flash | 0.053 | 0.084 | **+11.8%** | 3.26 | 22.5K | $0.003 |
| GPT-4o | 0.053 | 0.081 | **+11.8%** | 3.92 | 23.1K | $0.08 |

the computational overhead associated with local inference, with a cost of $0.003 per invocation, which is only 1/27 of GPT-4o, while achieving comparable downstream gains. At the same time, open-weight multimodal models in the 7B/3B range also exhibit competitive capability. Although a small performance gap remains, these models are still efficient and practical options.

Overall, these findings make Gemini-2.0-flash a natural choice as our standard policy model, offering the best balance between effectiveness and practical deployment, while it remains fully compatible with open-weight alternatives. This further demonstrates that ContextNav does not depend on any specific proprietary model and is robust and broadly applicable with respect to policy selection.

## D.4 TRADE-OFF BETWEEN ICL GAINS AND SYSTEM COSTS FOR DOWNSTREAM MLLMS

We applied the standard ContextNav implementation to Phi-3.5V, Qwen2.5-VL, Gemini-2.0-flash, and GPT-4o on BlindTest, and compared their ICL gains against token cost and latency (measured on MathVision). For latency, we report the system-level overhead introduced by ContextNav (averaged over multiple optimization iterations), excluding the downstream MLLM's own inference cost. The results in Table 10 show that ContextNav incurs broadly similar overhead across all models, while stronger downstream MLLMs exhibit slightly lower latency and token cost. We attribute this to their more accurate feedback, which facilitates smoother self-optimization within ContextNav. The results show that ContextNav could complete reasoning, optimization and actions within a short time (less than 5s) using only a small number of tokens (around 30K), while achieving strong downstream gains, which further highlights its efficiency and practical deployment feasibility.

## E  EXTENDED DISCUSSION

### E.1  DATASET SELECTION

In selecting our datasets, we aimed to ensure broad task diversity and strong representativeness. For composite-task datasets, we included: (1) MME-RealWorld: challenging comprehensive real-world multimodal understanding (such as complex scene and object recognition, ultra-high-resolution remote sensing, charts/tables in natural scenes, traffic perception, high-resolution surveillance). (2) BlindTest: basic visual tasks that are trivial for humans but difficult for MLLMs (e.g., counting intersections, circle intersection, overlapping shapes, nested squares, grid estimation). (3) CharXiv: natural scientific charts from real papers, capturing scientific multimodal QA in realistic settings. (4) GVL: topological multimodal reasoning tasks(connectivity, cycles, topological sort, shortest path, max flow, bipartite matching, Hamiltonian path). (5) MathVision: comprehensive multimodal mathematical reasoning tasks (algebra, geometry, combinatorics, statistics, topology, etc.). For single-task datasets, we follow the representative prior work VL-ICL, including CLEVR (attribute reasoning), FOMI (visual concept substitution), and TextOCR (optical character recognition).

Table 10: ICL gains, system latency, and token costs across downstream MLLMs.

| Downstream MLLM | ICL Gains ↑ | System Latency (s) ↓ | Token Costs ↓ |
|---|---|---|---|
| Phi-3.5V | 10.2% | 5.45 | 45.9K |
| Qwen2.5-VL-7B | 14.0% | 5.10 | 41.4K |
| Gemini-2.0-flash | 12.2% | 4.83 | 35.1K |
| GPT-4o | 10.3% | 4.68 | 33.3K |

### E.2 THE CONSTRUCTION PROCESS AND ROBUSTNESS OF OGG

The construction and edge-definition process of the OGG is automated. We instruct GPT-4o to infer tool composability by understanding and matching the input–output data formats of all atomic tools. Since each tool function has clearly defined I/O specifications, the automatically constructed OGG is highly reliable (as shown in Table 2, the toolchain execution success rate is close to 100%). Regarding ContextNav's robustness to the OGG design, we provide analyses from two complementary perspectives: database management and contextualization (downstream ICL gain).

**Robustness of database management to OGG design.** Because database management involves transforming unstructured corpus data into structured representations, the data-flow dependencies are strict. Missing, redundant, or incorrectly ordered tools can directly result in invalid data flows. To address this, we implement engineering-level validation: if a generated toolchain triggers a data-flow exception due to unsupported or ill-formed tool sequences, we automatically regenerate the chain. Only when repeated regeneration fails do we check whether the issue arises from the structure of the OGG; if so, we apply minimal manual adjustments to correct it.

**Robustness of downstream gains to OGG design.** The downstream ICL improvements primarily depend on the contexts produced during contextualization. To ensure the frameworks' robustness with respect to OGG design, we standardize the data-flow interface for contextualization tools, allowing them to be freely composed and enabling the agent to optimize their usage through memory and feedback. In practice, these tools are fully connected within the OGG (as shown in the appendix E), ensuring that contextualization never encounters missing or redundant tool dependencies, and thus does not negatively impact downstream ICL gains.

### E.3 ANALYSIS OF SYSTEM FAILURES

We observe a few special failure cases in the system. Specifically, in rare instances, using a smaller model (e.g., Qwen2.5-VL-3B) as the policy may cause it to repeatedly emit the toolchain in its output. This is an occasional token-level repetition loop that small models can exhibit in structured output spaces, where the model fails to distinguish between the toolchain-generation phase and the natural-language reasoning phase. As a result, the formatted sequence may be repeatedly produced and the EOS token may not appear. This phenomenon is rare and is only observed when we conduct ablation experiments with smaller policy models. It reflects a general model-level behavior rather than a system error with our framework, and therefore does not compromise the system's robustness. When this occurs, simply prompting the policy to regenerate the toolchain could effectively resolve the issue.

### E.4 LOOKING FORWARD: REINFORCEMENT LEARNING FOR AGENTIC MULTIMODAL ICL

We observe that the environment constructed by ContextNav is naturally suitable for RL. The agent's state can be directly instantiated from the retrieval process (such as the levels of semantic and structural noise), the context candidates (including their semantic and structural alignment with the query), and the memory module (which stores downstream ICL feedback and the history of previously executed toolchains). In addition, ContextNav provides a structured action space defined by the Operational Grammar Graph, which makes the environment's state transitions fully observable. Each selected action, such as generating a candidate set through retrieval, refining the candidates' semantic properties or noise profile, or updating the toolchain and downstream feedback, deterministically leads to a new state.

These components together form an RL trajectory in which the agent executes a sequence of interpretable operations, observes the resulting state updates, and accumulates feedback throughout the contextualization process. Furthermore, correctness signals from downstream feedback, the policy's reasoning cost (in terms of token usage), the retrieval noise levels and the candidate alignment levels can be jointly incorporated into a reward function that encourages improved downstream ICL performance while penalizing excessive inference token (fewer tokens help reduce inference latency) and overly noisy contextualization behavior. Such a formulation would enable the agent to learn more efficient and task-adaptive orchestration strategies over time without requiring substantial modifications to the system architecture.

While this line of investigation lies beyond the scope of the present work, we view RL-based policy optimization as a natural and compelling extension of ContextNav. We believe that RL is poised to play an important role in the next stage of agentic multimodal ICL, and our proposed ContextNav offers the first referenceable environment design that can meaningfully support this direction.

## F PROMPTS

---

**Tool Orchestration Prompt Template**

You are an agent responsible for retrieving information relevant to the user's query and integrating it into contextual knowledge to assist a multimodal large language model with in-context learning.

Your available tools are defined as functions with the following descriptions: {tool_library}. From the tool graph {textualized_tool_graph}, you must select one appropriate toolchain to automate the in-context learning process.

The following are the toolchain(s) you selected in previous steps together with the feedback received for your provided contextual knowledge: {memory}.

{system_constraints}.

Based on your reasoning, decide on the most appropriate toolchain at this step. You **must** first present your reasoning process, and then output your final decision strictly in the format: 'Toolchain: tool A -> tool B -> $\cdots$ -> tool N.', where the period "." marks the end of the output and **must not** be omitted.

---

**System Constraints Template**

There are some criteria you must follow: {criteria_and_retrieval-specification_Prompt}.

Please reason these questions and tell me your reasoning results: {chain_of_thought}.

---

**User-defined Criteria and Retrieval-specification Prompt**

1. If you are explicitly instructed that this is your **first step**, you **must** select a toolchain that contains the tools `textual_similarity_retrieval` and `visual_similarity_retrieval`. However, you cannot stop at these two tools; the complete toolchain must be specified.

2. If you are not explicitly told that this is your first step, or if you know it is not your first step (e.g., you already selected a toolchain in the previous step), you may select other toolchains at this step.

3. You must avoid re-selecting any toolchains that have already been chosen in previous steps.

4. If **all** toolchains have already been selected in previous steps, then you must disregard the above criteria and instead select a toolchain that includes at least the tools `textual_similarity_retrieval`, `visual_similarity_retrieval`, and `agentic_retrieval`.

---

**Chain-of-Thought**

1. Is the current step your **first step**?

2. If it is not your first step, list the toolchains you have already used in the previous steps.

3. Reflect on the feedback you received regarding the retrieved context information. Do you think the issues described in the feedback are related to the toolchains you selected in earlier steps?

---

---

**General Prompt for Agentic Retrieval**

<image_query> 

The two images above, together with the following questions, form two image–question pairs.

Question 1: {query_question}

Question 2: {ref_question}

You don't need to answer the questions. Just decide whether the two pairs share any similarity, either in the images (content) or in the question types (e.g., both ask for counting, scene understanding, etc.).

- If there is any similarity, reply: 'Judgement-YES'.

- If there is no similarity, reply: 'Judgement-NO' and briefly explain why.

(Optional: The similarity criterion does not need to be strict, any reasonable overlap counts as similarity.)

---

**General Prompt for Structural Alignment**

Rewrite the following question in the style of {query_question}.

Only output the rewritten question, without any explanations or extra text.

Question to rewrite: {ref_question}.

---

**Embedding Specification Prompt**

From the following options:

- Text models: {text_emb_model_zoo_prompt}

- Visual models: {vis_emb_model_zoo_prompt}

Select one text model and one visual model based on the hardware status {hardware_status}, considering disk and GPU memory usage base on {resource_usage_preference}.

**Output format:**

> Text Embedding: text_model_id;
> Image Embedding: visual_model_id

**Restriction:** Do not generate any additional symbols (e.g., **). If a vector database already exists, it is essential to ensure that the chosen embedding model is compatible with it.

---

**Default General In-Context Learning Template**

<image 1>...<image n> <image_query> (Option A)

I will provide a series of reference images, each paired with a corresponding question and answer. Your task is to **reflect on these references and summarize the useful information they convey**. After all references have been presented, I will then provide one final image with its question. Based on your prior reflections, you should give an answer to this final query.

The $k$-th reference sample is as follows (repeated for $n$ times):

- **Image $k$:** <image $k$> (Option B)
- **Question:** {ref_question[$k$]}
- **Answer:** {ref_answer[$k$]}

Finally, the last query is:

- **Final Image:** <image_query> (Option B)
- **Final Question:** {query_question}

Please use your summarized reflections from the reference samples to answer the final question.

{feedback_request}

*Note: Both option A and option B are acceptable for injecting images. For models with structured interfaces, text and images can be interleaved in a list format. However, when using open-source models, it is empirically*

---

> *common to place all image tokens at the beginning of the prompt, followed by textual instructions. In such cases, if images need to be dynamically inserted within the prompt, a recommended practice is to inject the contexts across multiple dialogue turns.*

**Feedback Request Prompt**

Please evaluate whether the reference samples you received were helpful and sufficiently rich (with the number of shots approximately matching the preset value {k}) in solving your final problem. If they were, additionally output "Judgement-Yes"; if not, additionally output "Judgement-No" and, starting with "Feedback:", explain whether the mismatch arose from the text or the image of the reference samples.

# G  TOOL GRAPH

## G.1  TOOL LIBRARY

| Atom Tools | Description |
| --- | --- |
| get_query | Receives the multimodal query (text and image) and initializes the workflow. |
| get_hardware_status | Monitors computational resources (e.g., GPU memory, disk capacity) to guide resource-aware embedding model selection. |
| check_updating | Detects newly added or modified samples in the context corpus and triggers re-embedding to ensure database synchronization. |
| matching_embedding_models | Selects appropriate text and vision embedding models from the model zoo, balancing retrieval quality with hardware efficiency. |
| multimodal_embedding | Converts multimodal corpus into vector representations, forming the basis for retrieval in the vector database. |
| load_vector_database | Builds and/or loads the multimodal vector database. |
| textual_similarity_retrieval | Retrieves semantically relevant candidates using text embeddings. |
| visual_similarity_retrieval | Retrieves visually correlated candidates using vision embeddings. |
| agentic_retrieval | Refines the initially retrieved candidates by filtering irrelevant or misleading examples through the agent's reasoning, mitigating semantic noise. |
| structural_alignment | Reorganizes textual structures of retrieved candidates, reducing structural noise and improving consistency with the query. |

Table 11: Descriptions of atom tools in the tool library.

## G.2  OPERATIONAL GRAMMER GRAPH

```
1  OGG_edges = [
2  ("start", "get_query"),
3  ("get_query", "get_hardware_status"),
4  ("get_query", "check_updating"),
5  ("get_query", "load_vector_database"),
6  ("check_updating", "get_hardware_status"),
7  ("check_updating", "multimodal_embedding"),
8  ("check_updating", "load_vector_database"),
9  ("get_hardware_status", "matching_embedding_models"),
10 ("matching_embedding_models", "multimodal_embedding"),
11 ("multimodal_embedding", "load_vector_database"),
12 ("load_vector_database", "textual_similarity_retrieval"),
13 ("load_vector_database", "visual_similarity_retrieval"),
```

```
14  ("textual_similarity_retrieval", "visual_similarity_retrieval"),
15  ("textual_similarity_retrieval", "agentic_retrieval"),
16  ("textual_similarity_retrieval", "structural_alignment"),
17  ("visual_similarity_retrieval", "textual_similarity_retrieval"),
18  ("visual_similarity_retrieval", "agentic_retrieval"),
19  ("visual_similarity_retrieval", "structural_alignment"),
20  ("agentic_retrieval", "structural_alignment"),
21  ("textual_similarity_retrieval", "end"),
22  ("visual_similarity_retrieval", "end"),
23  ("agentic_retrieval", "end"),
24  ("structural_alignment", "end")
25  ]
```

## H  DEFAULT MODEL ZOO

```
1   text_embedding_model_zoo = [
2   {"model_id": "Qwen/Qwen3-Embedding-8B",
3           "description": "Requires 18 GB of disk space and at least 32 GB of available
        ↪  GPU memory."},
4
5   {"model_id": "Qwen/Qwen3-Embedding-4B",
6           "description": "Requires 9 GB of disk space and at least 16 GB of available
        ↪  GPU memory."},
7
8   {"model_id": "Qwen/Qwen3-Embedding-0.6B",
9           "description": "Requires 2 GB of disk space and at least 8 GB of available
        ↪  GPU memory."},
10
11  {"model_id": "openai/clip-vit-large-patch14",
12          "description": "Requires 2 GB of disk space and at least 8 GB of available
        ↪  GPU memory."}
13  ]
```

```
1   vis_embedding_model_zoo = [
2   {"model_id": "Qwen/Qwen2.5-VL-3B-Instruct",
3           "description": "Requires 8 GB of disk space and at least 4 GB of available
        ↪  GPU memory (for ViT)."},
4   {"model_id": "openai/clip-vit-large-patch14",
5           "description": "Requires 2 GB of disk space and at least 8 GB of available
        ↪  GPU memory."}
6   ]
```

## I  CASE STUDY

### I.1  SUCCESSFUL CASES

We present in Figure 6 and Figure 7 a comparative case study using Gemini-2.0-flash as the target MLLM, illustrating the differences between ContextNav and alternative approaches in contextualization shot selection and the resulting ICL performance. As the figures demonstrate, ContextNav more effectively identifies semantically relevant contextual shots and aligns their textual structures more closely with the query, thereby enabling the MLLM to conduct ICL more reliably and generate more accurate responses to the input query.

### I.2  FAILURE CASES

We illustrate in Figure 8 and Figure 9 representative failure cases of ContextNav with Gemini-2.0-flash as the target MLLM. While ContextNav is generally effective at identifying and supplying the

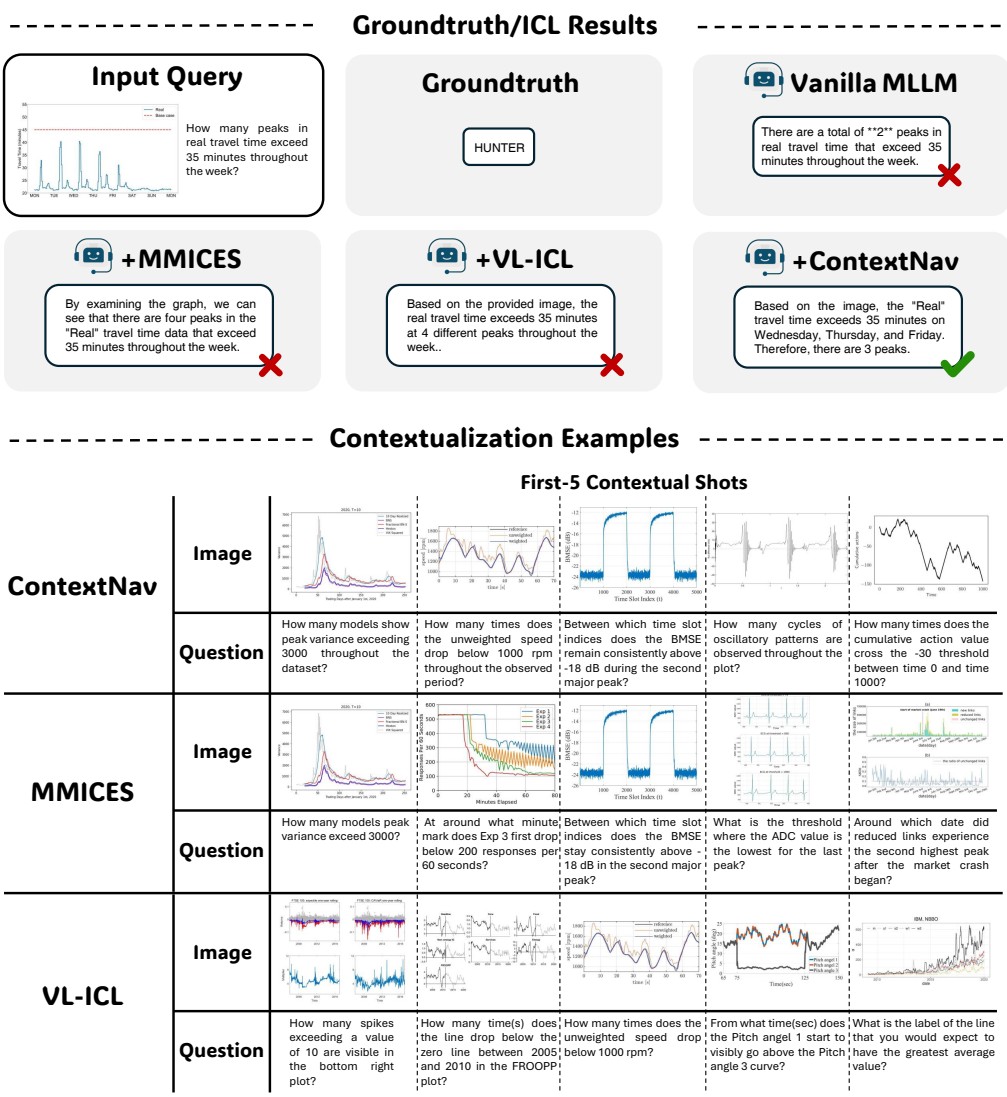

Figure 6: Example successful case from CharXiv.

target model with relevant contextual shots, and the model is often able to summarize and reflect upon these contexts, the final predictions are not always correct. We view such cases as reflecting the current limitations of downstream MLLMs' multimodal reasoning capabilities, which are difficult for ContextNav to overcome.

Figure 7: Example successful case from MathVision.

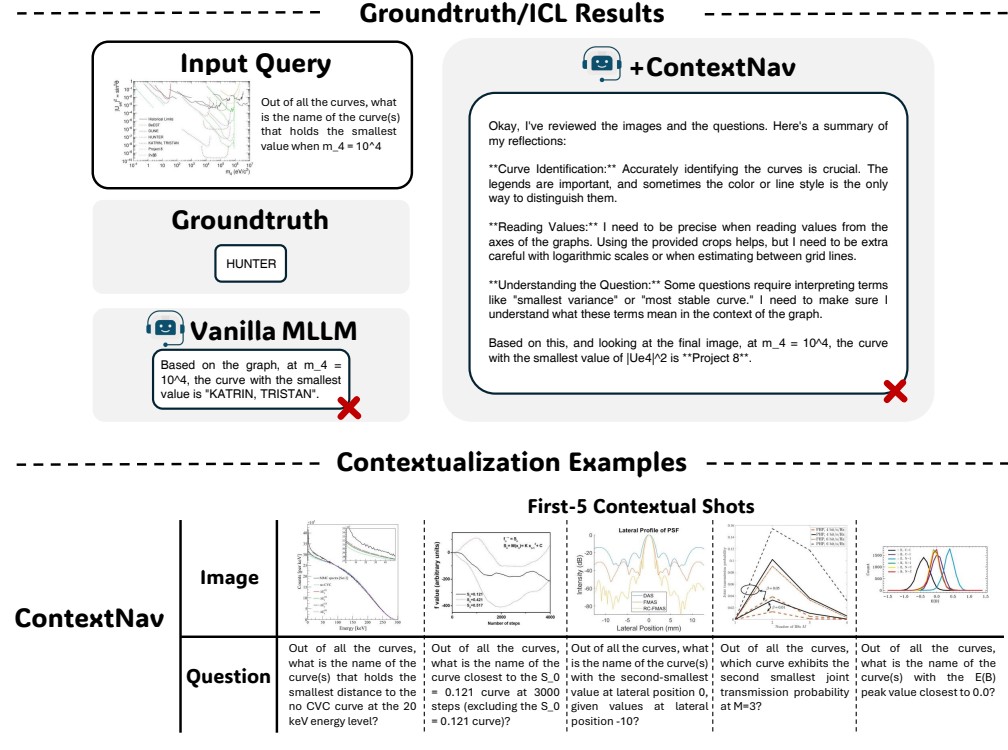

Figure 8: Example failure case from CharXiv.

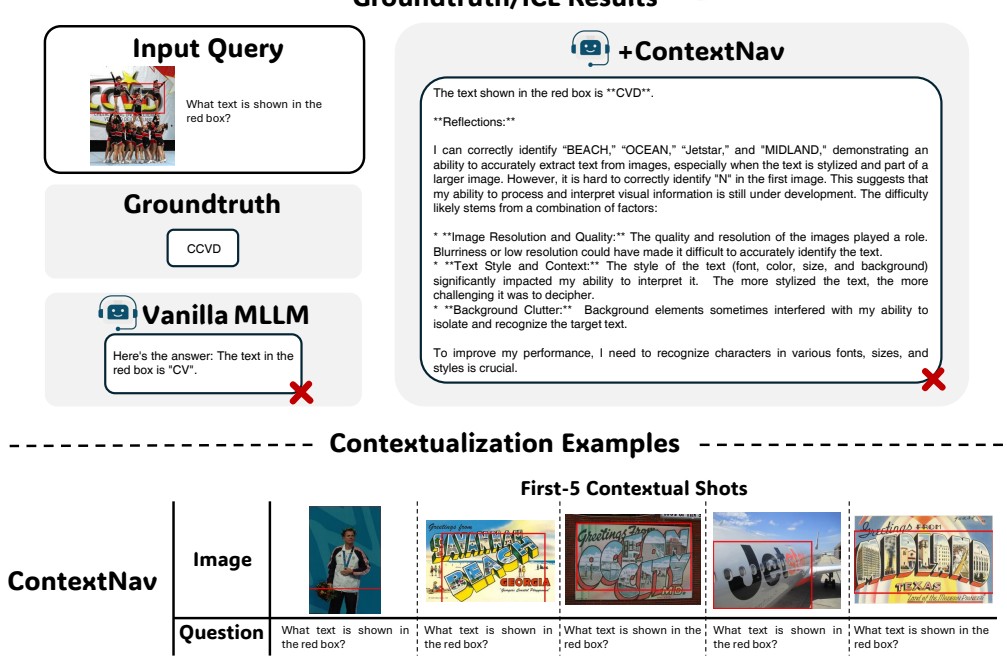

Figure 9: Example failure case from TextOCR.

