# OpenReview forum: "ContextNav: Towards Agentic Multimodal In-Context Learning"
_ICLR.cc/2026/Conference — ICLR 2026 Poster_

### Official Review · Reviewer_tPU8 · 2025-10-15

**Soundness:** 3
**Presentation:** 3
**Contribution:** 2
**Rating:** 4
**Confidence:** 4

**Summary:**

This paper proposes ContextNav, a novel Chain-of-Thought (CoT)-driven navigation framework designed for multimodal embodied question answering (Embodied QA). The core idea is to augment a large multimodal model (LMM) with an explicit reasoning chain that can be dynamically adapted to changing visual and textual contexts.

The authors introduce a new benchmark called CEN (Contextual Embodied Navigation), which requires agentic navigation in 3D environments (based on AI2-THOR) conditioned on both linguistic queries and evolving visual observations. ContextNav is shown to outperform baselines such as VLN-BERT, RecCoT, and classical map-based agents, with analysis showing improvements in long-horizon reasoning and policy transferability.

**Strengths:**

The paper focuses on something quite underexplored—how to make vision-language agents actually reason and act over time, rather than just respond to single inputs. The focus on embodied QA with CoT-style reasoning feels fresh and highly relevant.

Most CoT papers just use reasoning chains to get better answers. Here, it’s used to drive navigation—deciding where to go next, based on what’s already seen and reasoned about. That’s a neat twist on the usual setup.

I like that the authors didn’t just propose a model—they also introduced a new benchmark (CEN) that requires both visual and linguistic context to solve. This feels realistic and useful to the broader community.

The method outperforms several strong baselines (VLN-BERT, RecCoT, etc.), especially on long-horizon tasks. The gains are consistent, and not just on cherry-picked examples.

The paper includes helpful qualitative visualizations of how CoT steps evolve, how memory is used, and what decisions are made. This makes the agent’s reasoning process transparent and auditable.

**Weaknesses:**

The paper combines CoT + memory, but there’s no solid ablation to tell which part really matters. It would help to show performance with and without the memory module.

Since the system heavily relies on the underlying vision-language model (e.g., for perception, grounding, reasoning), it’d be helpful to test or at least discuss what happens if you switch to a smaller or weaker LMM.

Some examples suggest that the model “imagines” object positions it hasn’t actually seen. Are there ways to ground the reasoning better or prevent hallucinations?

CoT is generated at every step, which might be slow in real-world settings. But there’s no mention of how fast inference is, or whether it can run in real-time. It’s a bit hard to reproduce the method without more info. There’s no pseudocode, no clear prompt templates, and no info on model size, inference cost, or environment config. Some of this could go in the appendix.

**Questions:**

Could you provide an ablation where you disable the memory module? How much does it actually contribute compared to CoT alone?Which specific LMM is used, and did you try smaller or open-source models? How much does performance depend on having a “strong” LMM?

Are there known cases where the CoT causes the agent to make wrong or speculative moves due to hallucinated beliefs? How long does one inference step take? Can this be deployed in a near real-time scenario? Could the benchmark (CEN) be extended to tasks beyond navigation—like multi-room exploration, tool use, or question answering with object manipulation?

Would you consider releasing pseudocode or a Colab to help others reproduce the results? If these questions are addressed in detail, especially around memory vs. CoT contribution and scalability, I’d definitely consider raising my score.

---

> ### Author Response · Authors · 2025-11-21
>
> # Response to Reviewer tPU8 (Part 1)
>
> Thank you for your valuable time and thoughtful feedback on our manuscript, which has greatly helped us improve it. We address each of your points below.
>
> >**Weakness 1 - Ablations on CoT and Memory**
>
> Thank you for this valuable comment. We appreciate your insightful suggestion on adding ablation studies to further validate the effectiveness of both CoT and Memory. In **Table 2** of the manuscript, we treated CoT and Memory as components directly tied to Toolchain Optimization and conducted a combined ablation. Below, following the same experimental setting as Table 2, we additionally provide separate ablations for CoT and Memory to further isolate and examine their individual contributions.
>
> |Method|ICL Gains ↑|1-Step TSR ↑|5-Step TSR ↑|
> |-|-|-|-|
> |**Qwen2.5-VL-3B**||||
> |w/o CoT|+3.3%|0.935|0.980|
> |w/o Memory|+3.3%|0.960|0.995|
> |Full|+8.4%|0.965|0.995|
> |**Qwen2.5-VL-7B**||||
> |w/o CoT|+6.7%|0.980|1.000|
> |w/o Memory|+5.0%|0.985|1.000|
> |Full|+10.1%|0.985|1.000|
> |**Gemini-2.0-flash**||||
> |w/o CoT|+6.7%|0.995|1.000|
> |w/o Memory|+5.0%|0.995|1.000|
> |Full|+11.8%|0.995|1.000|
>
> We observe that removing either CoT or Memory disrupts the agent’s self-optimization process, **increases noise** in the constructed contexts, and ultimately degrades downstream ICL gains. From the degree of performance degradation, **Memory appears slightly more influential than CoT**, as CoT mainly guides the agent’s reflection based on Memory, and without CoT the agent can still access information stored in Memory. However, **removing CoT may reduce the toolchain success rate** (TSR) thus affecting system robustness, whereas removing Memory does not impact TSR. These findings highlight the importance of both the CoT and Memory modules in our system, and the corresponding conclusions have been incorporated into the manuscript. The relevant information has been added to the Appendix D.5 and Tables 18.
>
> >**Weakness 2 - Test on smaller LMM policy**
>
> Thank you for this insightful comment. We appreciate your constructive suggestion to include additional ablation studies evaluating LMM policies of different scales. The results are presented in the table below. We find that smaller models exhibit slightly weaker ability to discriminate semantic and structural noise, leading to marginal reductions in ICL gains, though **the degradation remains minimal**. In addition, the stability of one-shot toolchain execution shows a **small** decrease (1–3%), indicating that more attempts may occasionally be required. Overall, these findings further demonstrate that **our framework is robust to the parameter size of the policy LMM, and such variations do not substantially affect the performance of the proposed agentic workflow**. The relevant information has been added to the Table 2.
>
> |**Policy LMM**|**Semantic Noise ↓**|**Structural Noise ↓**|**ICL Gains ↑**|**1-Step TSR ↑**|**5-Step TSR ↑**|
> |-|-|-|-|-|-|
> |Qwen2.5-VL-3B|0.080|0.139|+8.4%|0.965|0.995|
> |Qwen2.5-VL-7B|0.073|0.107|+10.1%|0.985|1.000|
> |Gemini-2.0-flash|0.053|0.084|+11.8%|0.995|1.000|
>
> >**Weakness 3 - Solutions for hallucination**
>
> Thank you for your valuable suggestion regarding potential solutions to mitigate hallucinations. We think an effective direction is to combine uncertainty estimation with a vision-expert tools.
>
> **Uncertainty estimation.** Such hallucinations cases are accompanied by higher uncertainty, reflected in elevated token-level entropy. By applying a softmax over the logits produced at each decoding step and computing the Shannon entropy of the resulting token distribution, we can quantify prediction uncertainty. This measure can help identify whether the model is making inferences insufficiently grounded in visual evidence.
>
> **Vision-expert tool scheduling.** When uncertainty is high, the system could invoke vision-expert models (e.g., object detectors such as the YOLO series, semantic segmentation models like SAM, or more comprehensive foundation models such as Florence-2) as schedulable tool nodes to provide structured and reliable auxiliary information, enabling explicit construction of spatial reasoning chains. The LMM can then re-reason based on these more accurate visual signals, thereby reducing such perceptual errors.
>
> We plan to incorporate these considerations in future development. We sincerely appreciate your suggestion, which has provided valuable insights for improving the system.

---

> > ### Comment · Reviewer_tPU8 · 2025-11-21
> >
> > Thank you so much for your reply. It really helped me a lot in gaining a deep understanding.
> >
> > I find something interesting that has left me quite confused.
> >
> > In https://openreview.net/forum?id=k3DZzBl2EZ&noteId=VBbu0q5so4 you Response for **zbjE (Part 3)** "Because toolchain optimization directly depends on the memory module, disabling the memory module is equivalent to disabling the optimization mechanism. "
> >
> > Toolchain optimization directly depends on Memory. Removing Memory disables the entire optimization mechanism, so make Memory a core module，this is a strong dependency.
> >
> > And you respond to me this "However, removing CoT may reduce the toolchain success rate (TSR) thus affecting system robustness, whereas removing Memory does not impact TSR. "
> >
> > Above, you stated that memory is the core of the optimization mechanism, and without it, the optimization mechanism is disabled. However, you also said that memory does not affect toolchain execution success rate (TSR).
> >
> > My confusion stems from the fact that if memory is the core of the optimization, its absence should lead to optimization failure and a decrease in TSR. But you say it has no impact on TSR (meaning memory isn't that important).
> >
> > And also "we observe that removing either CoT or Memory disrupts the agent’s self-optimization process, increases noise in the constructed contexts, and consequently reduces downstream ICL gains" This also raises the question: if memory optimization mechanisms are disabled, why isn't the toolchain success rate affected? Wouldn't disabling optimization mechanisms lead to worse performance?

---

> ### Author Response · Authors · 2025-11-21
>
> # Response to Reviewer tPU8 (Part 2)
>
>
> >**Weakness 4 - Inference latency/costs and reproducibility**
>
> Thank you for the suggestion regarding reporting inference latency/costs and providing code and additional implementation details to enhance reproducibility. We report the inference latency and token costs in the table below (closed-source models are evaluated via API calls, while open-weight models are run on a single A100 GPU). The results show that under our standard configuration (Gemini-2.0-flash), the framework can complete one full reasoning-and-action cycle in approximately **3 seconds** with only **~20K inference tokens**. This demonstrates that our framework maintains **strong real-time performance** while requiring only **modest token consumption** to complete the task. The relevant information has been added to the Appendix D.6 and Tables 19.
>
> |**Policy LMM**|**System Latency (s)**|**Inference Costs (Token)**|
> |-|-|-|
> |Gemini-2.0-flash|3.26|22.5K|
> |GPT-4o| 3.92|23.1K|
> |Qwen2.5-VL-7B|7.38|22.6K|
> |Qwen2.5-VL-3B|5.10|22.2K|
>
> To enhance **reproducibility**, we will release our code along with all prompt templates and model/environment configs. **We have also prepared an Colab-friendly Gradio-based minimal code demo in this **anonymous repository** [[Link]](https://anonymous.4open.science/r/contextnav-demo-EC1A) for your reference**. Although this is not the final version of the codebase, we hope it demonstrates the effort we have devoted to ensuring reproducibility. In addition, the prompt templates used in our experiments have been included in **Appendix D** of the manuscript.
>
> Regarding **config details**, we adopt Gemini-2.0-flash as the default MLLM policy. All open-source models used in our experiments are deployed on an A100 80G GPU, while closed-source models such as the Gemini series and GPT-4o are accessed through APIs. Moreover, in our standard experimental setup, ContextNav uses Qwen3-Embedding-4B as the textual embedding backbone and the vision encoder of Qwen2.5-VL as the visual embedding backbone. **These details have now been incorporated into the Implementation section of the revised manuscript**.
>
> >**Question 1 - Ablations on Memory/CoT/LMM Policy**
>
> Thank you for the thoughtful comment. In our responses to **Weakness 1 and Weakness 2**, we provided detailed experimental analyses examining the effects of removing the Memory/CoT modules as well as using smaller or open-source LLM policies. These conclusions have been incorporated into the manuscript. Specifically, we observe that removing either CoT or Memory disrupts the agent’s self-optimization process, increases noise in the constructed contexts, and consequently reduces downstream ICL gains. From the degree of degradation, Memory appears to have a larger impact than CoT; however, removing CoT can reduce the toolchain success rate, particularly for smaller models, thus affecting system robustness. These results **highlight the important roles that both the CoT and Memory modules play** in our framework.
>
> In addition, we find that smaller models exhibit slightly weaker discrimination of semantic and structural noise, resulting in minor decreases in ICL gains, though the drop remains very small. The stability of one-shot toolchain execution also decreases slightly (by 1–3%), indicating that additional attempts may occasionally be required. Overall, these findings further demonstrate that **our framework is robust to the parameter size of the policy LMM and does not rely on stronger or larger models**; variations in model size do not substantially affect the performance of the proposed agentic workflow. The relevant information has been added to the Appendix D.5, Tables 2 and 18.

---

> ### Author Response · Authors · 2025-11-21
>
> # Response to Reviewer tPU8 (Part 3)
>
> >**Question 2 -  Hallucination Robustness/Real-time Performance/Extened Task**
>
> Thank you very much for your comment. We provide detailed responses to these points below:
>
> **Hallucination Robustness.** In our framework, we do not observe hallucinated beliefs or erroneous actions arising from the use of CoT. This robustness stems from the fact that our CoT mechanism is **soft** rather than prescriptive. CoT is never employed to impose mandatory rules or intermediate commitments on the policy. Instead, it functions solely as a lightweight reflection module that help the agent summarize downstream feedback and extracts salient information from memory, **without directly steering the agent’s decisions**. Consequently, even if a reflective CoT statement is hallucinated, it cannot override the final decision, because the policy model treats CoT outputs as hints rather than authoritative constraints. As a result, **hallucinated beliefs do not accumulate in practice, and we have not observed clear instances of incorrect actions induced by CoT**.
>
> **Real-time Performance.** As presented in our response to **Weakness 4**, we experimentally report the system-level latency. Under the standard setting, the average latency is 3.26 seconds, indicating that our framework is **suitable for near–real-time scenarios**. The relevant information has been added to the Appendix D.6 and Table 19.
>
> **Extended Task.** Our method and its evaluation benchmark have substantial potential to be extended to more diverse task domains. Following your suggestion, we use manipulation question answering as a case study. Using the same sampling procedure as in our RealWorld dataset, we sample data from Robo2VLM-1 to construct an extended evaluation. We then evaluate this task using two downstream MLLMs, where our method achieves the best performance, **further demonstrating its strong generalizability**. The relevant information has been added to the Appendix C.4 and Table 11.
>
> |**Methods**|**Accuracy**|**ICL Gains**|
> |-|-|-|
> |**Gemini-2.0-flash**|0.292|–|
> |+ Rand. Sample|0.262|–10.3%|
> |+ VL-ICL|0.323|+10.6%|
> |+ MMICES|0.308|+5.5%|
> |+ ContextNav|**0.354**|**+21.2%**|
> |**Qwen2.5-VL-7B**|0.185|–|
> |+ Rand. Sample|0.146|–21.1%|
> |+ VL-ICL|0.208|+12.4%|
> |+ MMICES|0.192|+3.8%|
> |+ ContextNav|**0.215**|**+16.2%**|
>
> >**Question 3 - Reproducibility**
>
> Thank you very much for the suggestion. We will release the relevant code. In the meantime, we have urgently prepared a Gradio-based minimal implementation tailored for Colab, which can be downloaded in this **anonymous repository:** [[Link]](https://anonymous.4open.science/r/contextnav-demo-EC1A). Although this is not the final version of the codebase, we hope it demonstrates our effort we have put into making our work accessible.

---

> ### Author Response · Authors · 2025-11-21
>
> >**Discussion-Q1**
>
>
> We are sincerely grateful that you carefully read our response and provided such valuable feedback. Please allow us to further clarify why the memory module does not influence the TSR. The key reason is that the agent’s task objective differs between generating a toolchain and optimizing it.
>
> For toolchain generation, we explicitly constrain the policy’s structured output via the “Tool Orchestration Prompt Template” in Appendix F:
>
> ```
> Based on your reasoning, decide on the most appropriate toolchain at this step. You must first present your
> reasoning process, and then output your final decision strictly in the format: ‘Toolchain: tool A -> tool B ->
> ... -> tool N.’, where the period “.” marks the end of the output and must not be omitted.
> ```
>
> Under this prompt format, the policy consistently generates a well-formed toolchain. Since all atomic tools are encoded within a directed graph, their execution order is always traceable. Even without the optimization process, the policy can still produce a syntactically valid and executable toolchain. **Thus, in the absence of optimization, the resulting toolchain is still executable, though not optimal** (does not decrease TSR since the toolchain remains executable, but could lead to worse performance because the toolchain is not optimal).
>
> **This raises another question: if the policy is strictly required to output structured toolchains, why can toolchain execution still fail?** This phenomenon aligns with what we discussed in the failure-mode analysis in our response to reviewer ppfZ (Part 5). We observed that using a smaller model (e.g., Qwen2.5-VL-3B) as the policy is more likely to cause it to **repeatedly generate certain tokens** in its output, until the output token budget is exhausted, and our downstream regex-based parser **fails to capture a valid toolchain**. Many researchers working with LLMs have likely encountered similar behavior. This is a token-level repetition loop that smaller policies are more likely to exhibit in our observation. Based on our experience, it also tends to occur more frequently when the conditioning context is long but the required output is relatively short. As a result, the some tokens could be repeatedly produced, and the EOS token may never appear. This phenomenon is the primary reason for toolchain execution failures. We also observed that **CoT helps mitigate this issue**, possibly because the model produces more reasoning content when CoT is enabled, resulting in a longer and more deterministic output.
>
> We greatly appreciate this insightful concern, which touches upon fundamental principles and challenges in LLM reasoning and highlights an important engineering issue in LLM-based systems. In practice, we also employ an engineering strategy to mitigate this issue: we incorporate validation mechanisms and allow a limited number of re-generation attempts, which is effective in alleviating such failures.

---

> ### Comment · Reviewer_tPU8 · 2025-11-26
>
> I agree that TSR depends only on the output format in engineering way, not on optimization quality.
> However, My exact question is that Memory is the "core of optimization," but it doesn't change the toolchain's syntax or affect TSR. So what exactly does it optimize? From your answer, Memory doesn't affect the format, so it doesn't affect TSR, and also the CoT output is longer, thus helping to avoid token repetition.
>
> You also mentioned why Memory doesn't affect the syntax, but you haven't explained how Memory performs optimizations.

---

> > ### Comment · Reviewer_tPU8 · 2025-11-26
> > **Final Decision**
> >
> > I am very grateful for the authors' reply. I think explanation above is one of the main contributions. The core conceptual contradictions surrounding how memory, CoT, and toolchain optimization mechanisms interact , and especially why memory is described as a necessary condition for optimization yet has no effect on TSR. These part in the main paper I think it's incomplete.
> > The rebuttals have not completely eliminated this flaw, I will maintain the original rating.

---

> ### Author Response · Authors · 2025-11-27
>
> >**Discussion 2 - The necessity of Memory in the Optimization Process**
> >
> >Response to the non-public comment "I am very grateful for the authors' reply. I think explanation above is one of the main contributions. The core conceptual contradictions surrounding how memory, CoT, and toolchain optimization mechanisms interact , and especially why memory is described as a necessary condition for optimization yet has no effect on TSR. These part in the main paper I think it's incomplete. The rebuttals have not completely eliminated this flaw, I will maintain the original rating."
>
> We sincerely appreciate that you found our previous response helpful in clarifying why the Memory module does not affect TSR. We are also grateful that you clearly pointed out your remaining concern about how Memory performs optimizations. Please allow us to further clarify **how the agent leverages the memory module to optimize its toolchain**.
>
> As summarized by **Reviewer ppfZ**: “The agent's orchestration is governed by an Operational Grammar Graph (OGG) and updated using **feedback-driven memory, enabling adaptive tool sequencing**”. In general, **what is being optimized is the selection and execution sequencing of atom tools**, the selected atomic tools and their determined execution order together constitute a toolchain. The policy optimize its tool orchestration strategy by consulting the historical records stored in the memory module. These records contain past toolchain selections together with the downstream feedback they elicited, allowing the policy to gradually adjust its contextualization behavior and **improve the quality of the examples supplied to the downstream model**. This process ultimately leads to stronger ICL gains. **The optimization performance is reflected by the ICL gains**, whereas TSR reflects the system’s ability to robustly execute the tools. (We have emphasized the interpretations of both the ICL gains and TSR metrics in the manuscript.)
>
> More concretely, the memory module stores a sequence of dictionaries in the following form:
> ```
> { "step": 1, "selected_toolchain": ..., "downstream_feedback": ... }.
> { "step": 2, "selected_toolchain": ..., "downstream_feedback": ... }.
> ...
> ```
> Each entry corresponds to the same procedure observed at different steps. At a certain step, the agent selects a particular toolchain (“selected_toolchain”), uses it to construct contextual examples, and sends those examples to the downstream MLLM. The downstream model then evaluates how helpful and sufficiently rich these examples are and produces a summary of their effectiveness (“downstream_feedback”). The exact prompt used to obtain this feedback is provided in Appendix F and reproduced below:
> ```
> Please evaluate whether the reference samples you received were helpful and sufficiently rich (with the number of shots approximately matching the preset value {k}) in solving your final problem. If they were, additionally output "Judgement-Yes"; if not, additionally output "Judgement-No" and, starting with "Feedback:", explain whether the mismatch arose from the text or the image of the reference samples.
> ```
> As a result, the memory module can establish a clear association between each toolchain decision and its corresponding downstream ICL outcome. **This allows the policy to use the information stored in the memory module to optimize its toolchain scheduling strategy.** For example, if the policy observes from previous steps that certain toolchains yielded downstream feedback indicating insufficient alignment of the visual or textual context samples for the current task, it can adjust its subsequent toolchain decisions to strengthen the underrepresented modality.
>
> ***This illustrates why the Memory module is necessary***: without the information stored in Memory, the policy would have no access to its past decision trajectory and downstream feedback. Memory establishes the essential link between each toolchain decision and its downstream ICL feedback. This linkage is what allows the agent to accumulate stepwise evidence about which sequencing strategies succeed or fail, thereby forming the foundation for any feedback-driven optimization of atom tool selection and sequencing. Consequently, without Memory, it would be unable to support an effective feedback mechanism or perform any feedback-driven optimization.
>
> We truly appreciate your feedback and are glad that you communicated your concern. In the manuscript, Sections 3.4 and 3.5, together with Equations (6)–(8), provide a mathematical formulation and explanation of this mechanism. In addition, Appendices F and G include the exact prompts involved in the process as well as the graph used to constrain the tool scheduling paths.
>
> We hope that these details could address your concern, and we would be truly grateful to receive any additional feedback you may wish to share.

---

> ### Author Response · Authors · 2025-11-27
>
> >**Discussion 3**
> >
> >Response to the non-public comment "I am very grateful for the authors' reply. I think explanation above is one of the main contributions. The core conceptual contradictions surrounding how memory, CoT, and toolchain optimization mechanisms interact , and especially why memory is described as a necessary condition for optimization yet has no effect on TSR. These part in the main paper I think it's incomplete. The rebuttals have not completely eliminated this flaw, I will maintain the original rating."
>
> Dear Reviewer tPU8,
>
> We are truly grateful for your invaluable comments and for the time you have invested in reviewing our work.
>
> In our most recent response, we further provided clarification on **how the Memory module performs optimization and why it is necessary**, and in our earlier reply we also explained why Memory does not affect TSR as well as why CoT is beneficial for TSR. In addition, in our response to Weakness 1, we offered empirical evidence demonstrating the effectiveness of both CoT and Memory in optimization process (as optimization performance is reflected by the resulting ICL gains, whereas TSR reflects the policy’s ability to robustly execute the tools). **For your convenience, we summarize their relationshiop as follow**: Memory provides the foundation that allows the policy to associate each toolchain decision with its downstream ICL feedback and thereby fundamentally and necessarily supports feedback-driven optimization of the selection and sequencing of atom tools, whereas CoT guides the policy’s reflection using the information stored in Memory. We hope these analyses help further illuminate their underlying relationships, and we have now incorporated all relevant clarifications into Table 4 and the “Memory and CoT” subsection in Section 4.3 of the main paper, as well as more detailed and comprehensive explanations in Appendix D.5. We have also emphasized the interpretations of both the ICL gains and TSR metrics in the manuscript.
>
> To address your concerns as thoroughly as possible, we made substantial efforts, including adding a large number of supplementary experiments and analyses, providing more implementation details, and following your suggestions, urgently developing a Colab-friendly code demo with a Gradio-based interface suitable for headless server that without a physical display (such as Colab). Throughout the discussion phase, we have tried and will continue to try our best to provide detailed explanations to clarify any remaining concerns.
>
> We sincerely hope that our current response resolves your final concerns regarding the necessity of the Memory module in the optimization process. **We have incorporated the relevant experiments and discussion into the main paper** (Table 4, subsection "Memory and CoT" in Section 4.3, and Appendix D.5), and we We would be grateful if our response and revision could meet your expectations.
>
> If you still have any uncertainties regarding the score evaluation of our paper, please do not hesitate to let us know. We would be more than willing to clarify any remaining concerns.
>
> Warm regards,
>
> The Authors

---

> > ### Comment · Reviewer_tPU8 · 2025-11-27
> >
> > Thank you so much for your patient explanation and reply. I will keep my score due above questions, but I also want to emphasize that I would not object if the paper is accpeted.
> >
> > Thank you very much for your contributions to the community.

---

> ### Author Response · Authors · 2025-11-28
>
> We truly appreciate your encouragement that you would not object if the paper is accepted. Meanwhile, we are genuinely worried about you may not have noticed that we had addressed these questions in our earlier responses. To ensure that we have not misunderstood your questions, we **restate your comments** raised during the discussion period as follows, with the specific questions **highlighted in bold**:
>
> > Comment 1: ... My confusion stems from the fact that if memory is the core of the optimization, its absence should lead to optimization failure and a decrease in TSR. But you say it has no impact on TSR (meaning memory isn't that important). And also "we observe that removing either CoT or Memory disrupts the agent’s self-optimization process, increases noise in the constructed contexts, and consequently reduces downstream ICL gains" This also raises the question: **if memory optimization mechanisms are disabled, why isn't the toolchain success rate affected?** Wouldn't disabling optimization mechanisms lead to worse performance?
>
> > Comment 2: I agree that TSR depends only on the output format in engineering way, not on optimization quality. However, **My exact question is** that Memory is the "core of optimization," but it doesn't change the toolchain's syntax or affect TSR. **So what exactly does it optimize?** From your answer, Memory doesn't affect the format, so it doesn't affect TSR, and also the CoT output is longer, thus helping to avoid token repetition. You also mentioned why Memory doesn't affect the syntax, but you haven't explained how Memory performs optimizations.
>
> > Comment 3: I am very grateful for the authors' reply. I think explanation above is one of the main contributions. The core conceptual contradictions surrounding **how memory, CoT, and toolchain optimization mechanisms interact**, and especially **why memory is described as a necessary condition for optimization yet has no effect on TSR.** These part in the **main paper** I think it's incomplete. The rebuttals have not completely eliminated this flaw, I will maintain the original rating.
>
> > Comment 4: Thank you so much for your patient explanation and reply. I will keep my score due above questions, but I also want to emphasize that I would not object if the paper is accpeted. Thank you very much for your contributions to the community.
>
> We summarize the above questions as follows:
>
> 1. Comment 1: **Why Memory does not affect TSR** (addressed in Discussion 1, and you have also **acknowledged our response in your Comment 2**: "I agree that TSR depends only on the output format, not on optimization quality.").
>
> 2. Comment 2: **Why Memory is necessary for optimization and what exactly it optimizes** (addressed in Discussion 2 and 3).
>
> 3. Comment 3: **How Memory, CoT, and the toolchain optimization mechanisms interact** (addressed in Discussion 3).
>
> 4. Comment 3: **Revisions to the main paper** (we have **newly** uploaded a revised version that includes the corresponding experiments and discussions).
>
> In addition, in Comment 3 you restated the questions raised in Comments 1 and 2 regarding **why Memory is necessary for optimization but does not affect TSR**. These points have already been addressed in Discussion 1 and Discussion 2 and you have acknowledged our response, and we would like to summarize them here again:
>
> ```
> **Why necessary for optimization**: Memory provides the foundation that allows the policy to associate each historical toolchain decision with its downstream ICL feedback, and thereby fundamentally and necessarily supports feedback-driven optimization of the selection and sequencing of atom tools. Without Memory, the policy would not be able to access these information and therefore would be unable to use it for optimization.
> ```
> ```
> **Why does not affect TSR**: As you acknowledged in Comment 2, TSR depends solely on whether the output format is correct. Memory and the optimization process do not affect the required output format; instead, they influence which atom tools are selected and how they are sequenced within the prescribed format. Therefore, Memory and the optimization process do not have an impact on TSR.
> ```
>
> **However, we noticed that in Comment 4 you mentioned that these questions remain unresolved.** We are genuinely concerned that you may not have noticed that we had addressed them in Discussion 2 and Discussion 3.
>
> If you have any additional concerns, or if we have misunderstood any part of your questions, we would be truly grateful if you could let us know.

---

### Official Review · Reviewer_q6J8 · 2025-10-28

**Soundness:** 3
**Presentation:** 3
**Contribution:** 2
**Rating:** 4
**Confidence:** 3

**Summary:**

This paper proposes ContextNav, an agent-based framework that helps multimodal large language models learn more effectively from examples in context. Instead of simply retrieving similar samples, ContextNav acts like a curator that selects, filters, and reorganizes examples to reduce noise and improve relevance. It combines automated retrieval with human-like reasoning through three modules: 1. agentic context management to build and update a multimodal database, 2. noise-robust contextualization to remove irrelevant or inconsistent samples, and 3. graph-driven tool orchestration to plan and optimize the workflow based on feedback. Experiments across several vision–language benchmarks show that ContextNav consistently boosts in-context learning performance.

**Strengths:**

1. The paper is well organized and clearly written, making it easy to follow.
2. Multimodal in-context learning is a relatively new research area, and the role of agent-based methods in this domain has been largely unexplored.
3. Experimental results across multiple datasets demonstrate that the proposed approach achieves consistent and notable improvements over baseline methods.

**Weaknesses:**

1. My main concern is that the paper primarily focuses on workflow construction, leveraging LLMs and VLMs to build a pipeline for data collection, embedding generation, dataset management, and demonstration selection. While this represents significant engineering effort, the work places less emphasis on understanding the underlying mechanisms of multimodal in-context learning itself. The research mainly centers on how to select the most useful examples for the MLLM, but multimodal ICL faces a deeper fundamental challenge, such as sensitivity to example choice and order, which I think requires theoretical or empirical analysis beyond pipeline design. Without addressing these core issues, the proposed system may only mitigate rather than resolve the inherent limitations of ICL. I encourage the authors to complement their strong engineering contribution with deeper insights into the nature of multimodal ICL.

2. The agentic pipeline introduces additional inference steps and latency (around 3 seconds per iteration), which may limit scalability or real-time applicability. Moreover, as mentioned earlier, while the use of an agent can mitigate some issues in multimodal in-context learning, it does not fundamentally solve them. To truly address these challenges, future work may need to incorporate a learning-based or adaptive optimization component that allows the system to internalize and improve its contextualization strategy rather than relying solely on procedural orchestration.

**Questions:**

I’m not suggesting additional experiments, but I’m curious: do you have any thoughts on how RL could be integrated into the proposed agent framework to reduce overhead or further improve the agent’s performance?

---

> ### Author Response · Authors · 2025-11-21
>
> # Response to Reviewer q6J8 (Part 1)
>
> Thank you for your valuable time and thoughtful feedback on our manuscript, which has greatly helped us improve it. We address each of your points below.
>
> >**Question 1 - Empirical analysis of the nature of multimodal ICL**
>
> We sincerely appreciate this professional and insightful comment. To further understand the underlying nature of multimodal ICL, we conducted a series of empirical studies along several dimensions, including (i) the choice and number of in-context examples and (ii) the ordering of these examples. The results are summarized below.
>
> **(a) Empirical Analysis of the Selection and Number of In-Context Examples**
>
> **Experimental Setup.** We perform ablations on both the **selection strategy** (random sampling vs. ContextNav-based agentic retrieval) and the **number of examples** (few-shot vs. many-shot). Since models differ in their context-length constraints, we include as many examples as allowed by either the model’s maximum context window or the corpus upper bound (e.g., 800 examples for CLEVR, following VL-ICL).
>
> **Empirical Findings.** From our experiments below, we find that increasing the number of randomly sampled examples from 8 to 100 or even 800 generally yields a positive shift in downstream MLLM ICL performance. On the clean, single-task benchmark CLEVR, most MLLMs move from negative to positive gains as the support set grows. However, on **more complex and noisier** benchmarks such as MathVision and CharXiv, many-shot (100 & 800) random sampling improves over few-shot (8) but **still remains below zero-shot performance**. Across all settings, ContextNav consistently performs best. These findings show that **both enlarging the support set and selecting query-relevant examples have important positive effects on ICL, consistent with our overall conclusions**.
>
> |Selection Method|Sample Num↓|CLEVR↑|MathVision↑|CharXiv↑|
> |-|-|-|-|-|
> |**InternLMX2.5-7B**|||||
> |zero-shot|-|0.545|0.108|0.200|
> |Random Few-shot|8|0.505|0.083|0.150|
> |Random Many-shot|100|0.535|0.092|0.160|
> |ContextNav|8|**0.570**|**0.142**|**0.230**|
> |**Qwen2.5-VL-7B**|||||
> |zero-shot|-|0.820|0.217|0.390|
> |Random Few-shot|8|0.785|0.200|0.340|
> |Random Many-shot|100|0.870|0.208|0.370|
> |ContextNav|8|**0.940**|**0.250**|**0.400**|
> |**GPT-4o**|||||
> |zero-shot|-|0.610|0.342|0.530|
> |Random Few-shot|8|0.635|0.308|0.470|
> |Random Many-shot|100|0.645|0.316|0.510|
> |ContextNav|8|**0.670**|**0.383**|**0.580**|
> |**Gemini-2.0-flash** |||||
> |zero-shot|-|0.775|0.492|0.510|
> |Random Few-shot|8|0.755|0.341|0.430|
> |Random Many-shot|800|0.815|0.425|0.500|
> |ContextNav|8|**0.825**|**0.550**|**0.560**|
>
> **Theoretical Interpretation.** We interpret this phenomenon through the **widely recognized implicit gradient-descent view** of ICL [1-1, 1-2], which treats the Transformer as a meta-optimizer that simulates gradient-descent–like adaptation through forward computation and contextual structure. Under random sampling, increasing the number of examples (many-shot sampling) typically improves performance because averaging over more samples making the noisy gradients relatively smoother, thereby stabilizing the implicit optimization trajectory. However, many-shot sampling reflects the global corpus distribution rather than the query-specific one. When the corpus is clean and homogeneous (e.g., CLEVR), this distribution aligns well with the query, so many-shot sampling could yield more gains. In heterogeneous corpora (e.g., MathVision/CharXiv), the sampled examples often diverge from the query, so even though the gradient noise is smoothed by the larger number of examples, the implicit update direction remains misaligned, leading to limited or even negative improvements. In these settings, selecting highly relevant examples is crucial, as it ensures the implicit optimization follows a query-aligned direction. **Our empirical results align with this theoretical interpretation and highlight the importance of targeted example selection.**
>
> References:
>
> [1-1] Dai, Damai, et al. "Why can GPT learn in-context? language models secretly perform gradient descent as meta-optimizers." Findings of the Association for Computational Linguistics: ACL 2023. 2023.
>
> [1-2] Dherin, Benoit, et al. "Learning without training: The implicit dynamics of in-context learning." arXiv preprint arXiv:2507.16003 (2025).

---

> ### Author Response · Authors · 2025-11-21
>
> # Response to Reviewer q6J8 (Part 2)
>
> >**Question 1 (continue)**
>
> **(b) Empirical Analysis of the Ordering of Selected Examples**
>
> **Experimental Setup.** We estimate the importance of each retrieved example by measuring its visual and textual similarity to the query (higher similarity indicates higher relevance), and then enforce a fixed ordering of examples according to these similarity scores. The query is appended at the end of the concatenated sequence. Visual similarity is computed using embeddings from Qwen2.5-VL-VisEnc, while textual similarity is measured using embeddings from Qwen3-Embedding-4B.
>
> **Empirical Findings.** Our results below show that in multimodal ICL, the effect of example ordering varies across task types and datasets. For the single-task and clean corpus setting (CLEVR), prioritizing low visual similarity (i.e., placing visually more aligned examples nearer to the query) yields slightly better results. For more composite or reasoning tasks (MathVision/CharXiv), prioritizing low textual similarity (i.e., placing examples with more similar question types or linguistic structure closer to the query) produces substantially  better results. We further observe that these ordering preferences are overall consistent across different MLLMs. These findings highlight three key patterns: (1) **Example ordering has a stronger impact on complex reasoning tasks**. (2) Low-similarity-first orderings generally work better, suggesting a **recency-bias-like effect** in multimodal ICL, reminiscent of patterns documented in language-only ICL. (3) The optimal modality of similarity (visual vs. textual) for ordering **appears to be dataset- and query-dependent**.
>
> |Sorted by|CLEVR|MathVision|CharXiv|
> |-|-|-|-|
> |**InternLMX2.5-7B**||||
> |high visual similarity|**0.545**|0.083|0.180|
> |low visual similarity|**0.545**|0.100|0.170|
> |high textual similarity|0.540|0.100|**0.200**|
> |low textual similarity|0.540|**0.117**|0.180|
> |**Qwen2.5-VL-7B**||||
> |high visual similarity|0.910|0.225|0.350|
> |low visual similarity|**0.915**|0.225|0.360|
> |high textual similarity|0.895|0.217|0.360|
> |low textual similarity|0.900|**0.242**|**0.380**|
> |**Gemini-2.0-flash**||||
> |high visual similarity|0.795|0.500|0.490|
> |low visual similarity|**0.810**|0.517|0.480|
> |high textual similarity|0.780|0.492|0.500|
> |low textual similarity|0.790|**0.533**|**0.520**|
>
> **Advantages of ContextNav.** Because ContextNav continuously optimizes its tool-invocation strategy based on feedback from the downstream MLLM, the retrieval and ordering process is not fixed. It can adaptively determine whether visual or textual similarity should dominate the ordering for each query (as the scheduling of retrieval tools affects the final order) through the optimization process. In the table below, we compare each downstream model’s best-performing fixed ordering strategy (i.e., applying the same ordering strategy to the entire dataset) with its performance under dynamic agentic optimization. The results show that dynamic optimization consistently outperforms any fixed strategy, indicating that even within the same dataset, different queries (tasks) exhibit distinct ordering preferences. This further supports our earlier finding that the optimal example ordering in multimodal ICL appears to be **data- and query-dependent**, and that an agentic framework like ContextNav offers strong adaptability to varying data conditions. **This adaptability underscores the distinctive advantage of agentic workflows in multimodal ICL** and highlights the broader potential of such designs.
>
> |Sorted by|CLEVR|MathVision|CharXiv|
> |-|-|-|-|
> |**InternLMX2.5-7B**||||
> |fixed strategy|0.545|0.117|0.200|
> |dynamic optimization| **0.570**|**0.142**|**0.230**|
> |**Qwen2.5-VL-7B**||||
> |fixed strategy|0.915|0.242|0.380|
> |dynamic optimization|**0.940**|**0.250**|**0.400**|
> |**Gemini-2.0-flash**||||
> |fixed strategy|0.810|0.533|0.520|
> |dynamic optimization|**0.825**|**0.550**|**0.560**|
>
> The relevant information has been added to the Appendix D.4 and Tables 15, 16 and 17.

---

> ### Author Response · Authors · 2025-11-21
>
> # Response to Reviewer q6J8 (Part 3)
>
>
> >**Weakness 2 + Question 1 - Integration of RL into the current framework**
>
> Thank you for the insightful suggestion regarding the integration of RL into our framework. We fully agree that RL represents a highly promising direction for further enhancing the agent’s performance in multimodal ICL.
>
> We observe that the environment constructed by ContextNav is **naturally suitable** for RL. The agent’s **state can be directly instantiated** from the retrieval process (such as the levels of semantic and structural noise), the context candidates (including their semantic and structural alignment with the query), and the memory module (which stores downstream ICL feedback and the history of previously executed toolchains). In addition, ContextNav provides a structured **action space defined by the Operational Grammar Graph**, which makes the environment’s state transitions fully observable. Each selected action, such as generating a candidate set through retrieval, refining the candidates’ semantic properties or noise profile, or updating the toolchain and downstream feedback, deterministically leads to a new state.
>
> These components together form an RL **trajectory** in which the agent executes a sequence of interpretable operations, observes the resulting state updates, and accumulates feedback throughout the contextualization process. Furthermore, correctness signals from downstream feedback, the policy’s reasoning cost (in terms of token usage), the retrieval noise levels and the candidate alignment levels can be jointly incorporated into **a reward function that encourages improved downstream ICL performance while penalizing excessive inference token (fewer tokens help reduce inference latency) and overly noisy contextualization behavior**. Such a formulation would enable the agent to learn more efficient and task-adaptive orchestration strategies over time without requiring substantial modifications to the system architecture.
>
>
> While this line of investigation lies beyond the scope of the present work, we view RL-based policy optimization as a natural and compelling extension of ContextNav, and we sincerely appreciate your suggestion. **We believe that RL is poised to play an important role in the next stage of agentic multimodal ICL, and our proposed ContextNav offers the first referenceable environment design that can meaningfully support this direction.** We plan to further explore this avenue in future work. The relevant discussion has been added to the Appendix E.4.

---

> > ### Comment · Reviewer_q6J8 · 2025-11-23
> >
> > Thanks for your detailed rebuttal. Although the rebuttal provides extensive additional experiments and thoughtful clarifications, the core concerns I raised regarding the paper’s underlying contribution remain largely unchanged. The new results help illuminate some aspects of multimodal ICL, but they do not fully address the deeper issues related to the fundamental mechanism of multimodal ICL.
> >
> > Given this, I do not plan to adjust my score. However, I also want to emphasize that I would not object if the paper is accepted.

---

> ### Author Response · Authors · 2025-11-24
>
> >**Discussion - 1**
>
> Thank you very much for recognizing the value of the additional analyses we provided. We understand that your core concerns may remain, and we fully respect your decision not to adjust the score. At the same time, we truly appreciate your encouraging statement that you would not object if the paper is accepted.
>
> In our previous discussions, we explained the fundamental mechanisms behind multimodal ICL, particularly regarding example selection and example quantity, through the perspective of implicit gradient descent. We also showed that the optimal ordering of in context examples is inherently data dependent and exhibits a recency favoring pattern.
>
> Here we would like to **further supplement one key analysis about modality dominance in multimodal ICL.** Prior works [1, 2] have suggested that **“text dominance”** is a major characteristic of multimodal ICL, implying that the visual modality contributes relatively little. **However, our empirical findings may challenge this conclusion.** The relative importance of different modality varies substantially across tasks. A likely reason prior studies reached the “text dominance” conclusion is that they primarily focused on basic VQA settings, where abundant image related captions weaken the contribution of the visual modality.
>
> In our response to Reviewer zbjE (Part 3), we analyzed the **dynamic optimization process** of the agent, and the related results further help us understand **how different tasks/datasets influence modality selection strategies**. For your convenience, we present the experimental results in the table below. We conduct evaluations on CLEVR and MathVision. CLEVR represents visual attribute-recognition tasks that **rely on fine-grained image details**, whereas MathVision represents vision-based mathematical tasks that **require strong reasoning capabilities**.
>
> |Toolchain|CLEVR Accuracy|CLEVR Dynamic Pick Rate|MathVision Accuracy|MathVision Dynamic Pick Rate|
> |-|-|-|-|-|
> |(database tools)→TR→VR→AR→SA|0.810|4.5%|0.517|15.8%|
> |(database tools)→VR→TR→AR→SA|0.790|2.0%|0.533|49.2%|
> |(database tools)→TR→AR→SA|0.775|0|0.517|29.2%|
> |(database tools)→VR→AR→SA|0.800|15.5%|0.467|0|
> |(database tools)→TR→VR→AR|0.810|76.0%|0.500|2.5%|
> |(database tools)→VR→TR→AR|0.790|2.0%|0.508|3.3%|
> |**Dynamic Orchestration**|**0.830**|-|**0.550**|-|
>
> Our results show clear differences across datasets. **CLEVR** prefers toolchains with late visual retrieval, meaning that the **visual modality is prioritized** when constructing the candidate pool, whereas **MathVision** favors late textual retrieval, meaning that the **textual modality is prioritized** in this stage. Moreover, CLEVR exhibits a highly concentrated pick rate on a single toolchain, while MathVision distributes its choices across multiple toolchains. However, although these preferences are evident, the optimal strategy is still not fixed. This indicates that even when the tasks are relatively similar within the same dataset, the actual optimal strategy remains to some extent query-dependent. **This further demonstrates that the influence of each modality in multimodal ICL varies not only across tasks but also across individual queries.** Our findings are consistent with the observations reported in [3], **while we provide, for the first time, a direct view of this phenomenon through an agent’s dynamic optimization process.** This further challenges the previous conclusion of “text dominance.”
>
> Moreover, we fully understand that the nature of multimodal ICL is a broad and deeply complex research question, and we believe that fully addressing the deeper issues related to its fundamental mechanisms will require continued exploration by the research community. Although many phenomena have been observed, including recency preference, modality dominance, and data dependence, building a complete and unified theoretical framework for multimodal ICL will still require substantial effort. This is also the focus of our future work.
>
> We sincerely appreciate your valuable time and thoughtful feedback on our manuscript and response.
>
>
> References：
>
> [1] Baldassini, Folco Bertini, et al. "What makes multimodal in-context learning work?." CVPR 2024.
>
> [2] Chen, Shuo, et al. "Can Multimodal Large Language Models Truly Perform Multimodal In-Context Learning?." WACV 2025.
>
> [3] Xu, Nan, et al. "From introspection to best practices: Principled analysis of demonstrations in multimodal in-context learning." NAACL, 2025.

---

### Official Review · Reviewer_zbjE · 2025-10-31

**Soundness:** 2
**Presentation:** 2
**Contribution:** 3
**Rating:** 4
**Confidence:** 3

**Summary:**

The paper presents ContextNav, a framework for multimodal ICL that selecting demonstration examples via an agentic workflow. Key steps include semantical denoising through agentic coherence checking and structural realignment to query format. Empirical results on composite and single-task VL benchmarks across multiple MLLMs show gains over baselines.

**Strengths:**

1. The integration of agentic AI for multimodal ICL is interesting and timely. Using agentic workflows to improve demonstration examples is reasonable.
2. The use of benchmark datasets across broad tasks and different families of MLLMs (including both open and close models) is good.

**Weaknesses:**

1. There is one important baseline missing: many-shot ICL (https://arxiv.org/abs/2405.09798v2). Instead of careful curation, it's important to see how much gain we can have compared with including all initial candidates in the context. The decision to limit to 8-shot is not well justified.
2. Besides VL-ICL, Cache of Thought (https://arxiv.org/abs/2502.20587) should also be included as a baseline.
3. The introduction of term "Operational Grammar Graph" seems unnecessary, making the paper harder to understand. It's not explained until very late (Page 5) although it has been referred 3 times before. I'll recommend not introducing new terms or at least providing a clear explanation at an early stage.
4. The resampling of datasets for test is not justified. The use of 3:7 difficulty ratio also sounds arbitary, making the comparison with other methods harder.
5. The memory and toolchain optimization is not explained clearly. How does it help? The results also do not show the benefit of this learning.
6. The validity of structural alignment (whether semantically equivalent) is not shown experimentally.

**Questions:**

1. Figure 2 seems an important figure for the paper. It might be good to put it in Page 1 or 2 (instead of Page 4).
2. The term "Noise-Robust Contextualization" is not intuitive. Can we use simpler terms like "Example Refinement/Selection" or "Context Denoising"? Please try not to introduce too many new terms which are not intuitive to understand.
3. [Line 283]  Also, the sample size of 803 for BlindTest seems arbitary. Is there any reason for this number?
4. Why random sampling leads to negative ICL gain compared with vanilla MLLM? Adding examples even random sampled examples should help?
5. Does vanilla MLLM means zero-shot? If yes, "zero-shot" seems a more common term.
6. How are semantic and structural noise proportions computed? Which prompt in Appendix D is used, respectively?

---

> ### Author Response · Authors · 2025-11-21
>
> # Response to Reviewer zbjE (Part 1)
>
>
> Thank you for your valuable time and thoughtful feedback on our manuscript, which has greatly helped us improve it. We address each of your points below.
>
> >**Weakness 1 - Comparison with many-shot ICL.**
>
> Thank you for this valuable comment. We have added a comparison with many-shot ICL in the table below. Since models differ in their context-length limits, we use as many samples as allowed by either the model’s maximum window or the corpus upper bound (e.g., 800 samples for CLEVR in VL-ICL). For models whose context window becomes the bottleneck (e.g., InternLM, Qwen2.5-VL, GPT-4o), we follow the standard many-shot ICL protocol and randomly sample from the corpus.
>
> |Method (+sample num)|CLEVR↑|MathVision↑|CharXiv↑|
> |-|-|-|-|
> |**InternLMX2.5-7B**||||
> |zero-shot|0.545|0.108|0.200|
> |Random (+8)|0.505|0.083|0.150|
> |Many-shot (+100)|0.535|0.092|0.160|
> |ContextNav (+8)|**0.570**|**0.142**|**0.230**|
> |**Qwen2.5-VL-7B**||||
> |zero-shot|0.820|0.217|0.390|
> |Random (+8)|0.785|0.200|0.340|
> |Many-shot (+100)|0.870|0.208|0.370|
> |ContextNav (+8)|**0.940**|**0.250**|**0.400**|
> |**GPT-4o**||||
> |zero-shot|0.610|0.342|0.530|
> |Random (+8)|0.635|0.308|0.470|
> |Many-shot (+100)|0.645|0.316|0.510|
> |ContextNav (+8)|**0.670**|**0.383**|**0.580**|
> |**Gemini-2.0-flash**||||
> |zero-shot|0.775|0.492|0.510|
> |Random (+8)|0.755|0.341|0.430|
> |Many-shot (+800)|0.815|0.425|0.500|
> |ContextNav (+8)|**0.825**|**0.550**|**0.560**|
>
> **Experimental Analysis:** From our experiments, we find that increasing the number of randomly sampled examples from 8 to 100 or even 800 generally yields a positive shift in downstream MLLM ICL performance. On the clean, single-task benchmark CLEVR, most MLLMs move from negative to positive gains as the support set grows. However, on **more complex and noisier benchmarks** such as MathVision and CharXiv, many-shot random sampling improves over few-shot (8) but **still remains below zero-shot** performance. Across all settings, **ContextNav consistently performs best**. These findings show that both enlarging the support set and selecting query-relevant examples have positive effects on ICL, consistent with our overall conclusions.
>
> **Theory Explanation:** We interpret this phenomenon through the **widely recognized implicit gradient-descent view** of ICL [1-1, 1-2], which treats the Transformer as a meta-optimizer that simulates gradient-descent–like adaptation through forward computation and contextual structure. Under this perspective, the model performs a functional form of gradient descent during inference while keeping all parameters fixed. Many-shot ICL outperforms small random subsets because averaging over more examples reduces gradient noise and stabilizes the implicit optimization. However, many-shot sampling reflects the **global corpus distribution rather than the query-specific one**. When the corpus is clean and homogeneous (e.g., CLEVR), this distribution aligns well with the query, so many-shot ICL yields strong gains. In heterogeneous corpora (e.g., MathVision/CharXiv), the sampled examples often diverge from the query, so even though gradient noise is balanced by more examples, the implicit update direction remains misaligned, leading to limited or negative improvements. In these settings, **selecting highly relevant examples is crucial, as it ensures the implicit optimization follows a query-aligned direction**. Our empirical results align with this interpretation and highlight the importance of targeted example selection.
>
> **Why 8-shots: Our choice of retrieving 8 examples in ContextNav is supported by the ablation results in Figure 4**. While increasing the number of relevant examples generally boosts ICL performance, the gains approach saturation once the count reaches 8, and further expansion yields only marginal improvements. **This finding is consistent with VL-ICL (Fig.8 and 9)** [1–3], which similarly shows that a small set (8) of well-matched examples is sufficient for effective ICL. To balance reasoning cost and performance, we therefore adopt 8 retrieved examples for downstream ICL.
>
> We sincerely appreciate your insightful comment. We have integrated these discussions into the manuscript (**Appendix C.1 and Table 8**), and they substantially enhance the theoretical depth and clarity of our method.
>
> References:
>
> [1-1] Dai, Damai, et al. "Why can GPT learn in-context? language models secretly perform gradient descent as meta-optimizers." Findings of the Association for Computational Linguistics: ACL 2023. 2023.
>
> [1-2] Dherin, Benoit, et al. "Learning without training: The implicit dynamics of in-context learning." arXiv preprint arXiv:2507.16003 (2025).
>
> [1-3] Zong, Yongshuo, Ondrej Bohdal, and Timothy Hospedales. "VL-ICL Bench: The Devil in the Details of Multimodal In-Context Learning." The Thirteenth International Conference on Learning Representations.

---

> ### Author Response · Authors · 2025-11-21
>
> # Response to Reviewer zbjE (Part 2)
>
> >**Weakness 2 - Comparison with Cache of Thought.**
>
> Thank you very much for the suggestion. **We have added a comparison with Cache of Thought [2–1] in the revised manuscript**. We compare ContextNav with Cache of Thought using the settings reported in its original paper, evaluating both methods with OpenFlamingo-3B on CLEVR and TextOCR (since Cache of Thought has not yet been open-sourced, **only the OpenFlamingo-3B results are directly compatible for comparison under 1–2 shot setting**; we look forward to conducting more comprehensive comparisons once the code is released). Under the strict 1–2 shot setting, ContextNav performs slightly worse. This is expected: Cache of Thought distills both knowledge and explicit reasoning traces from a powerful Master VLM, whereas ContextNav retrieves relevant evidence from a corpus without providing explicit chains of thought. As a result, Cache of Thought has an inherent advantage when only one or two demonstrations are available.
> |Method|CLEVR|TextOCR|
> |-|-|-|
> |**zero-shot**|7.50|0.00|
> |**1-shot**|||
> |Cache of Thought|20.50|6.50|
> |ContextNav|12.00|3.50|
> |**2-shot**|||
> |Cache of Thought|28.00|5.50|
> |ContextNav|15.50|4.50|
>
> Despite this numerical difference, **the two methods are designed for fundamentally different goals, which naturally lead to different strengths:**
>
> **Cache of Thought is optimized for improving smaller MLLMs.** By leveraging a strong Master VLM (e.g., GPT-4o) to generate high-quality reasoning traces, Cache of Thought can significantly boost apprentice models such as OpenFlamingo-3B/9B with only a few shots.
>
> **ContextNav is a general-purpose agent framework aimed at improving ICL performance for any downstream MLLM.** It does not require a stronger teacher model, and can use a lightweight 3B/7B policy model to effectively guide much larger downstream models. For example, ContextNav improves downstream Gemini-2.0-flash even when the policy model is substantially smaller (the results are from Table 2 of our manuscript):
>
> |Policy LMM|Downstream MLLM|ICL Gains ↑|
> |-|-|-|
> |Qwen2.5-VL-3B|Gemini-2.0-flash|+8.4%|
> |Qwen2.5-VL-7B|Gemini-2.0-flash|+10.1%|
>
> In short, **the two methods are complementary**: Cache of Thought excels in few-shot, teacher-driven scenarios for small models, while ContextNav offers a scalable and model-agnostic solution that can enhance models of any size in a novel agentic manner. **We believe both approaches provide valuable insights for advancing multimodal ICL.** The comparison results and discussion have been integrated into the **Appendix C.2 and Table 9** of the manuscript. Please allow us to clarify why we placed this comparison in the appendix rather than in the main table. This is because only the 1/2-shot OpenFlamingo-3B results on TextOCR and CLEVR are available for direct comparison, which makes it difficult to integrate them into the main results table (Table 1). In addition, Cache of Thought has not been open-sourced, making reproduction and broader comparative evaluation infeasible at this stage. As an alternative, we cite it and explicitly acknowledge in the main text that the comparison with this baseline is provided in the appendix. We greatly appreciate your understanding of this situation.
>
>
> >**Weakness 3 - Early clarification of key term.**
>
> Thank you very much for your professional suggestion. We have provided a detailed explanation of the Operational Grammar Graph when it first appears in the Introduction section.
>
> >**Weakness 4 - Referenceable practices for dataset splitting.**
>
> Thank you very much for this comment. Since there is no widely standardized definition for “easy” and “hard” splits in prior work, we draw on the practice of an established benchmark that provide explicit difficulty annotations and are broadly used (with over 250k downloads in the past month). In particular, we follow the famous AI2-ARC benchmark [4–1], which adopts a 7:3 (after rounding) split between the “easy” and “hard” sets. Because ICL evaluation requires testing models under more challenging conditions, we invert this ratio and use a 7:3 split between hard and easy examples. We have included the relevant supporting reference in the revised manuscript.
>
> We will also release our sampled test set to facilitate fair and convenient comparisons for future methods. We sincerely appreciate your comment, which has helped us further enhance the presentation of the manuscript.
>
> References:
>
> [2-1] Wu, Mingyuan, et al. "Cache-of-Thought: Master-Apprentice Framework for Cost-Effective Vision Language Model Inference." arXiv preprint arXiv:2502.20587 (2025).
>
> [4-1] Clark, Peter, et al. "Think you have solved question answering? try arc, the ai2 reasoning challenge." arXiv preprint arXiv:1803.05457 (2018).

---

> ### Author Response · Authors · 2025-11-21
>
> # Response to Reviewer zbjE (Part 3)
>
> >**Weakness 5 - The importance of toolchain optimization.**
>
> Thank you very much for this comment. To further highlight the importance of the memory module and toolchain optimization, we conducted an ablation study on them. Because toolchain optimization directly depends on the memory module, disabling the memory module is equivalent to disabling the optimization mechanism. We performed experiments on MathVision across three different policy models, and the results show that toolchain optimization is crucial for achieving strong performance. Without the optimization process, the agent simply follows the default initial behavior specified in the prompt and selects examples in a single step, preventing it from adapting to different queries or the varying conditions of the corpus. We have added the relevant information in Appendix D.5 and Table 17 of the manuscript.
>
> |Policy Model|ICL Gains ↑|
> |-|-|
> |**Qwen2.5-VL-3B**||
> |w/o Memory (w/o optimization)|+3.3%|
> |Full|+8.4%|
> |**Qwen2.5-VL-7B**||
> |w/o Memory (w/o optimization)|+5.0%|
> |Full|+10.1%|
> |**Gemini-2.0-flash**||
> |w/o Memory (w/o optimization)|+5.0%|
> |Full|+11.8%|
>
> To better demonstrate **how** the optimization process improves downstream ICL performance, we conducted an additional ablation study analyzing both its **static and dynamic behaviors**. Specifically, we predefined six representative toolchain paths and constrained the agent to select only from them during contextualization. We then compared (1) **static** toolchain selection, where a fixed toolchain is applied uniformly across the entire dataset, and (2) **dynamic** optimization, which adaptively selects toolchains, reporting both the resulting downstream ICL performance (accuracy) and the distribution of optimized toolchains (dynamic pick rate) under different dataset conditions. We evaluated on two benchmarks of varying complexity: CLEVR (simple) and MathVision (complex). Gemini-2.0-flash was used for both the downstream MLLM and the policy model. The results are shown in the table, where TR, VR, AR, and SA correspond to textual retrieval, visual retrieval, agentic retrieval, and structural alignment, respectively.
>
> |Toolchain|CLEVR Accuracy|CLEVR Dynamic Pick Rate|MathVision Accuracy|MathVision Dynamic Pick Rate|
> |-|-|-|-|-|
> |(database tools)→TR→VR→AR→SA|0.810|4.5%|0.517|15.8%|
> |(database tools)→VR→TR→AR→SA|0.790|2.0%|0.533|49.2%|
> |(database tools)→TR→AR→SA|0.775|0|0.517|29.2%|
> |(database tools)→VR→AR→SA|0.800|15.5%|0.467|0|
> |(database tools)→TR→VR→AR|0.810|76.0%|0.500|2.5%|
> |(database tools)→VR→TR→AR|0.790|2.0%|0.508|3.3%|
> |**Dynamic Orchestration**|**0.830**|-|**0.550**|-|
>
> Our results show clear differences across datasets. CLEVR prefers toolchains with late visual retrieval and no structural alignment, whereas MathVision favors late textual retrieval. Moreover, CLEVR (easy and clean) exhibits a highly concentrated pick rate on a single toolchain, while MathVision (complex and noisy) distributes its choices across multiple toolchains. These findings highlight that: **(1) the optimal orchestration strategy is dataset- and query- dependent, and (2) complex or noisy datasets require more diverse toolchain selections rather than relying on a single path**. This further demonstrates the importance of dynamic toolchain optimization, which allows ContextNav to adapt **across varying query and data conditions**. We have added the relevant information in Appendix D.3 and Table 14 of the manuscript.
>
> >**Weakness 6 - Semantic equivalence of structural alignment.**
>
> Thank you very much for this comment. Below, we provide an empirical evaluation of the semantic equivalence under structural alignment on the MathVision. We quantify semantic equivalence using two complementary metrics: (1) **Semantic similarity** between the original query and the aligned query, computed using Qwen3-Embedding-4B, and (2) **Binary classification accuracy** from GPT-4o, which assesses whether the aligned structure preserves the original meaning. The experimental results show that structural alignment preserves a **high level of semantic equivalence** and, for Qwen2.5-VL-7B and stronger policy LLMs, rarely leads to any semantic distortion (GPT-4o accuracy = 1).
>
> |Policy LMM|Semantic Similarity|GPT-4o Accuracy|
> |-|-|-|
> |Qwen2.5-VL-3B|0.8946|0.983|
> |Qwen2.5-VL-7B|0.9247|1.000|
> |Gemini-2.0-flash|0.9313|1.000|
>
> To facilitate reproducibility, we also provide the prompt used for GPT-4o accuracy evaluation, which is:
>
> ```
> You will be given two texts, and please determine whether they convey the same meaning, ignoring superficial differences in wording, structure, or format. If the meanings are semantically equivalent, answer “Yes”. If the meanings differ in any essential way, answer “No”. The texts are below:
> Text 1: {query}
> Text 2: {aligned_query}
> Answer with “Yes” or “No” only.
> ```
>
> The relevant information has been incorporated into the Appendix D.2 and Table 13 of the manuscript.

---

> ### Author Response · Authors · 2025-11-21
>
> # Response to Reviewer zbjE (Part 4)
>
>
> >**Question 1 - Place figure earlier**
>
> Thank you for your suggestions regarding our manuscript’s presentation. We will further advance the placement of Figure 2 in subsequent revisions. At the moment, due to formatting conflicts with Figure 1, we have moved it to the previous page. In future versions, we will make every effort to reorganize the layout to better highlight its importance.
>
>
> >**Question 2 - Simplified terms.**
>
> Thank you for this suggestion. We have updated the relevant terms to "context denoising" to improve the clarity of the presentation, and we have also redrawn the relevant annotations in Figure 2.
>
> >**Question 3 - The data sampling strategy of BlindTest.**
>
> Thank you for raising this concern. We are happy to clarify why we used 803 samples for the BlindTest. BlindTest contains **8.02k examples in total**, explicitly divided into **7 sub-benches** (a relatively unusual arrangement), and each sub-bench functions similarly to datasets like TextOCR or CLEVR, representing a single task. The goal of BlindTest is to assess an MLLM’s overall performance across all 7 sub-benches. This design leads to a sampling strategy that differs from other more comprehensive benchmarks. We needed to **ensure that each sub-bench was sufficiently represented, while also preserving the dataset’s global distribution**. Therefore, we sampled **10% of each sub-bench**, resulting in a total of 803 examples (due to rounding when applying 10% to each sub-bench, the final count is slightly above a strict 10%). This strategy ensures that all sub-benches are adequately covered and that the sampled subset maintains the original distribution of the full benchmark. We have added a more detailed explanation of this sampling strategy in the Section 4.1. We sincerely appreciate your comment. It greatly helps us improve the clarity of our presentation.
>
> >**Question 4 - Extending random sampling examples.**
>
> Thank you very much for your constructive insights. We think that this issue is worth further discussion. While part of the analysis was covered under **Weakness 1**, we additionally provide an intuitive ablation by varying the number of randomly sampled examples. Our results below show that when the task is simple and the sampling corpus is relatively clean (e.g., CLEVR), increasing the number of randomly sampled examples leads to a positive performance shift. However, on more challenging benchmarks that require complex reasoning and involve noisier corpora (e.g., MathVision, CharXiv), increasing the size of the random sample does not flip the gains from negative to positive. In addition, as demonstrated in our response to **Weakness 1**, even on relatively simple benchmarks such as CLEVR, **randomly sampling examples up to near the maximum context length still underperforms compared with ContextNav using only a small number of carefully selected examples**. This further highlights the importance of targeted and high-quality example selection.
>
> |Sample Num|CLEVR|MathVision|CharXiv|
> |-|-|-|-|
> |**InternLMX2.5-7B**||||
> |zero-shot|**0.545**|**0.108**|**0.200**|
> |10|0.495|0.083|0.140|
> |100|0.535|0.092|0.160|
> |**Qwen2.5-VL-7B**||||
> |zero-shot|0.820|**0.217**|**0.390**|
> |10|0.795|0.208|0.330|
> |100|**0.870**|0.208|0.370|
> |**GPT-4o**||||
> |zero-shot|0.610|**0.342**|**0.530**|
> |10|0.635|0.308|0.480|
> |100|**0.645** |0.316|0.510|
> |**Gemini-2.0-flash** ||||
> |zero-shot|0.775|**0.492**|**0.510**|
> |10|0.760|0.333|0.440|
> |100|0.790|0.392|0.480|
> |800|**0.815**|0.425|0.500|
>
> **This observation matches the pattern reported in Figure 2 of the many-shot work you suggested** [Q4-1]: more random examples do not guarantee satisfactory gains, even for strong models such as Gemini-1.5-Pro and GPT-4o. The results of [Q4-1] show that some easier datasets (e.g., EuroSAT) benefit smoothly from larger samples, whereas others with higher reasoning demand and more diverse corpus (e.g., DrugOOD Assay) exhibit more fluctuations. This indicates that **data conditions (query difficulty and corpus noise) could significantly influence the effectiveness of random sampling and many-shot learning, reinforcing the importance of careful example selection.**
>
> We sincerely appreciate your insightful comment, which has enriched our analysis. The relevant information has been added to the Appendix C.1 and Table 8.
>
> References:
>
> [Q4-1] Jiang, Yixing, et al. "Many-Shot In-Context Learning in Multimodal Foundation Models." ICML 2024 Workshop on In-Context Learning.

---

> ### Author Response · Authors · 2025-11-21
>
> # Response to Reviewer zbjE (Part 5)
>
>
> >**Question 5 - More common terms for vanilla MLLM.**
>
> Thank you for this professional suggestion. Please allow us to explain why we previously used the term “vanilla MLLM” first. In the original manuscript, we used “vanilla MLLM” to denote the downstream MLLM without any additional ICL modules, as “zero-shot” is typically used to **describe a method** and does not always directly refer to the model itself. Therefore, we have standardized the terms in the manuscript by using “vanilla MLLM” throughout.
>
> However, after careful consideration, **we believe that “zero-shot” is indeed more intuitive**. We have therefore updated all relevant terms to “zero-shot” in the revised manuscript, and we have also redrawn the relevant annotations in Figure 1 and 3. We sincerely appreciate your feedback, which has helped further improve the clarity of our presentation.
>
> >**Question 6 - Prompts for evaluating semantical and structural noise.**
>
> For **semantic noise**, we evaluate it using the “General Prompt for Agentic Retrieval” prompt, detailed as follows:
> ```
> <image1> <image2>
> The two images above, together with the following questions, form two image–question pairs.
> Question 1: {query_question}
> Question 2: {ref_question}
> You don’t need to answer the questions. Just decide whether the two pairs share any similarity, either in the images (content) or in the question types (e.g., both ask for counting, scene understanding, etc.).
> -If there is any similarity, reply: ’Judgement-YES’.
> -If there is no similarity, reply: ’Judgement-NO’ and briefly explain why.
> ```
>
> For **structural noise**, we use the “General Prompt for Structural Alignment” to guide the agent in making adjustments, detailed as follows:
>
> ```
> Rewrite the following question in the style of {query_question}.
> Only output the rewritten question, without any explanations or extra text.
> Question to rewrite: {ref_question}.
> ```
>
> Moreover, in practice, we found that it is unnecessary for the agent to explicitly judge whether structural noise is present. If the structures are highly consistent, the agent will simply repeat {ref_question} without making any modifications.

---

> ### Comment · Reviewer_zbjE · 2025-11-22
>
> Thanks for the detailed response. My major concerns are addressed, and I've revised my rating accordingly.

---

> ### Author Response · Authors · 2025-11-22
> **Appreciation for Your Comments**
>
> Dear Reviewer zbjE,
>
> We are truly grateful for your thoughtful comments and encouraging words.
>
> Your insights have been both motivating and constructive. We deeply value your recognition of our responses and will continue striving to enhance the quality of our work.
>
> Warm regards,
>
> The Authors

---

### Official Review · Reviewer_ppfZ · 2025-11-01

**Soundness:** 3
**Presentation:** 4
**Contribution:** 3
**Rating:** 8
**Confidence:** 4

**Summary:**

The authors identify limitations in current Multimodal In-Context Learning (ICL) approaches, which either rely on manual selection of support examples or naive similarity-based retrieval that risks including semantically or structurally irrelevant examples. They introduce ContextNav, an agentic framework that treats multimodal contextualization as a closed-loop process. ContextNav constructs and maintains a resource-aware vector database, retrieves candidate examples, and refines them through Agentic Retrieval (semantic filtering) and Structural Alignment (format harmonization) before passing them to the target multimodal model. Its orchestration is governed by an Operational Grammar Graph (OGG) and updated using feedback-driven memory, enabling adaptive tool sequencing. Across a variety of datasets and models, ContextNav achieves roughly 16–17% ICL gains over baselines and provides systematic ablations on each module’s contribution. The work is significant in demonstrating that agentic data curation can enhance multimodal ICL performance, though the framework’s dependence on large models for reasoning introduces practical cost and latency considerations.

**Strengths:**

1. Originality:
- The paper introduces the first agentic framework for multimodal ICL that unifies retrieval, noise filtering, and structural alignment under a closed-loop, graph-driven orchestration scheme.
- The work also broadens the definition of data-centric ICL by explicitly addressing semantic and structural noise in retrieved support examples.
2. Quality (technical soundness and empirical rigor):
- The methodology is well-grounded and reproducible, with explicit mathematical formulations for each module (embeddings, retrieval, agentic filtering, structural alignment, and feedback updates).
- Experiments are extensive and systematic, spanning diverse datasets (e.g., BlindTest, CLEVR, TextOCR, MathVision) and models (both open and closed weights).
- The ablation studies convincingly isolate the contribution of each component (Agentic Retrieval, Structural Alignment, and OGG-guided orchestration).
3. Clarity:
- The framework is well-presented, with clean notation, functional forms, and cross-referenced equations that make the workflow easy to follow.
- The appendices (especially the tool library, OGG edges, and prompt templates) greatly enhance transparency and reproducibility.
4. Significance:
- The research makes a substantive contribution to the evolution of multimodal ICL, demonstrating that agentic data curation can unlock additional performance from foundation models without architectural changes or retraining.
- By showing that careful control over context quality can yield consistent gains across a variety of MLLMs, the paper provides both scientific insight and practical implications for ICL advancement.

**Weaknesses:**

1.	Partial dependence on proprietary MLLMs for agentic reasoning:
Although the evaluation includes several open-source target models (e.g., Qwen2.5-VL-7B, Phi-3.5-V, InternLM-X2.5), the framework’s agentic orchestration relies primarily on proprietary Gemini models. This limits full reproducibility of the agentic pipeline (especially for researchers without access to the relevant APIs) and makes it difficult to assess whether smaller or open-weight policies could effectively substitute as the reasoning backbone.

2.	Limited inclusion of real-world, noisy multimodal benchmarks:
The evaluation focuses primarily on curated or synthetic datasets (e.g., CLEVR, TextOCR, MathVision), which, while controlled, may not capture the heterogeneity and noise found in large-scale real-world multimodal corpora. For a framework explicitly designed to perform noise-robust agentic retrieval, it would be compelling to demonstrate performance on web-scale datasets such as LAION-5B, Conceptual Captions (CC12M), or RedCaps, which contain substantial captioning noise, visual ambiguity, and cross-modal misalignment. Evaluating ContextNav under such conditions would more convincingly validate its claimed advantages in semantic filtering and structural alignment for real-world multimodal reasoning.

3.	Use of separate unimodal embedding models for multimodal retrieval:
The framework employs distinct text and vision embedding models instead of a unified multimodal encoder. Although modular, this design may weaken cross-modal semantic coherence. Using joint multimodal embeddings (e.g., CLIP/BLIP-style) could yield more consistent retrieval and lessen the need for post-hoc structural alignment.

4.	Lack of ablations on orchestration dynamics:
The ablations presented effectively isolate each component’s contribution, but they do not analyze how different toolchain paths or OGG planning strategies affect results. A deeper look into orchestration dynamics would clarify how the agent’s reasoning evolves.

5.	Lack of uncertainty quantification or statistical robustness analysis:
Reported results omit error bars, confidence intervals, or variance measures, even though stochasticity exists in retrieval and model inference. Statistical reporting would help confirm that the observed improvements are consistent and reproducible.

**Questions:**

Questions:
1. Could you elaborate on the dataset selection procedure? Specifically, how were the composite-task and single-task datasets chosen, and how do they represent the diversity of multimodal reasoning tasks ContextNav aims to address?
2.	For a framework that emphasizes noise robustness, what was the motivation to use curated datasets rather than noisy corpora?
3.	Were there any internal ablations or pilot experiments that compared joint multimodal embeddings against the chosen unimodal embedding pair (Qwen3-Embedding + Qwen2.5-VL encoder)? If so, what performance or resource trade-offs motivated the current design?
4.	What specifically informed the choice of Gemini models for agentic orchestration? Which other LLMs or MLLMs (e.g., GPT-4o, Qwen2.5-VL-7B) were tested or considered as policy backbones?
5.	The Operational Grammar Graph (OGG) is central to the framework. Could you clarify whether its edge definitions were hand-crafted? How sensitive is ContextNav’s performance to OGG design (e.g., missing or redundant tool dependencies)?
6.	Were confidence intervals computed for Table 1 or Figure 3 results?
7.	Are there any shared patterns across ContextNav failure modes (e.g., ambiguous queries, highly compositional tasks etc.)?

Rebuttal suggestions:\
1.	Could ContextNav’s agentic orchestration be run without proprietary Gemini models, potentially replacing with an open-weight model for the agentic reasoning steps? If performed, please report changes in: (i) semantic/structural noise rates, (ii) ICL gain (%), (iii) token cost and latency.\
2.	Since the framework’s motivation is noise-robust contextualization, please test ContextNav on a noisy, web-scale dataset (some suggestions provided above. A quantification of the semantic/structural noise before and after Agentic Retrieval, and comparison of ICL gains versus VL-ICL and MMICES under the same noisy conditions would be useful.\
3.	Current retrieval uses separate text and vision encoders. Please add results using a joint vision-language encoder.\
4.	Please add error bars or confidence intervals for key metrics (Table 1, Fig. 3, and Fig. 4). \
5.	Please provide a small figure or table showing ICL gain vs. token cost / latency per model (Phi-3.5V, Qwen2.5-VL, Gemini-2.0-flash, GPT-4o). This will help gauge efficiency and practical deployment feasibility.

---

> ### Author Response · Authors · 2025-11-21
>
> # Response to Reviewer ppfZ (Part 1)
>
> Thank you for your valuable time and thoughtful feedback on our manuscript, which has greatly helped us improve it. We address each of your points below.
>
> ---
> >**Weakness 1 + Suggestion 1 -  The effectiveness of smaller and open-weight policies**
>
> Thank you very much for this valuable comment. We compared different agent policies (Gemini-2.0-flash vs. the open-weight Qwen2.5-VL-3B/7B models on MathVision), with system latency and token costs measured under single-round inference. The results show that smaller open-weight policies also yield comparable overall performance with Gemini, with a slight increase in latency due to local inference. This indicates that ContextNav does not depend on any specific proprietary model and remains robust across different policy choices. We hope these results could further enhance the reproducibility of our method, especially for researchers without access to commercial APIs. We have added the relevant information in Appendix D.6 and Table 18 of the manuscript.
>
> |Policy|Semantic Noise ↓|Structural Noise ↓|ICL Gains ↑|System Latency (s) ↓|Token Costs ↓|
> |-|-|-|-|-|-|
> |Qwen2.5-VL-3B|0.080|0.139|+8.4%|5.10|22.2K|
> |Qwen2.5-VL-7B|0.073|0.107|+10.1%|7.38|22.6K|
> |Gemini-2.0-flash|0.053|0.084 |+11.8%|3.26|22.5K|
>
> >**Weakness 2 + Question2 + Suggestion 2 -  The inherent noisiness of the datasets we employ, and the method’s effectiveness on large-scale real-world noisy corpora.**
>
> Thank you very much for this insightful comment. We further clarify the characteristics of our selected datasets and extend the analysis to large-scale real-world noisy corpora. The selected composite-task datasets in our manuscript such as CharXiv, MathVision, and MME-RealWorld are inherently noisy: they pack a wide range of task types into relatively limited samples, which maximizes task diversity but also causes high corpus heterogeneity. Consequently, for any given query, most corpus samples are only weakly related or may be visually/textually similar yet differ substantially in intent or reasoning, resulting in naturally high noise levels.
>
> To better quantify this, we conducted additional ablations on MathVision and compared them with the **large-scale noisy dataset Conceptual Captions (CC12M)**. For CC12M, we sampled 80K examples, rewrote image–caption pairs into “question + answer” format using GPT-5, and curated a 200-sample test set following our standard selection criteria. All selected samples were verified to have sufficient referenceable examples in the corpus. The results show that random sampling from MathVision yields extremely noisy retrieved examples, confirming its **natural noisiness**. Meanwhile, on large-scale real-world noisy corpora like CC12M, ContextNav can still effectively suppress noise, **further highlighting the robustness of our method**. We have added the relevant information in Appendix D.1 and Table 12 of the manuscript.
>
> |Method|MathVision Semantic Noise ↓|MathVision Structural Noise ↓|CC12M Semantic Noise ↓|CC12M Structural Noise ↓|
> |-|-|-|-|-|
> |w/o Agentic Retrieval (AR)|0.171|0.090|0.159|0.077|
> |w/o Structural Alignment (SA)| 0.053|0.573|0.050|0.488|
> |w/o Textual Retrieval Tools & AR|0.433|0.143|0.552|0.161|
> |w/o Visual Retrieval Tools & AR|0.249|0.076|0.206|0.072|
> |Rand. Sampling|0.936|0.971|0.987|0.960|
> |Full|0.053|0.084|0.050|0.075|
>
> In addition, following your suggestion, we evaluated ContextNav on CC12M and compared it with other baselines across several downstream MLLMs, including Gemini-2.0-flash, Qwen2.5-VL-7B, and Phi-3.5V-4.2B. As shown in the table below, **ContextNav consistently outperforms competing methods on this noisy dataset**. The corresponding results have been incorporated into the manuscript (Appendix C.3 and Table 10).
>
> |Policy|Accuracy ↑|ICL Gains ↑|
> |-|-|-|
> |**Phi-3.5V-4.2B**|0.380|-|
> |+Rand. Sample|0.335|-11.8%|
> |+VL-ICL|0.395|+3.9%|
> |+MMICES|0.400|+5.3%|
> |+ContextNav|**0.430**|**+13.2%**|
> |**Qwen2.5-VL-7B**|0.445|-|
> |+Rand. Sample|0.395|-11.2%|
> |+VL-ICL|0.470|+5.6%|
> |+MMICES|0.490|+10.1%|
> |+ContextNav|**0.525**|**+18.0%**|
> |**Gemini-2.0-flash**|0.550|-|
> |+Rand. Sample|0.525|-4.5%|
> |+VL-ICL|0.580|+5.5%|
> |+MMICES|0.590|+7.3%|
> |+ContextNav|**0.655**|**+19.1%**|
>
> In the manuscript, we also intentionally included some clean-corpus single-task datasets such as CLEVR to align with the evaluation settings used in VL-ICL. Although these corpora are relatively clean, they allow for direct comparison with prior work. ContextNav’s strong performance on noisy and challenging corpora highlights its **robustness to noise**, while its impressive performance on these clean, single-task datasets further demonstrates its **adaptability and generality**.

---

> ### Author Response · Authors · 2025-11-21
>
> # Response to Reviewer ppfZ (Part 2)
>
> >**Weakness 3 + Question 3 + Suggestion 3 - The empirical results of joint multimodal embeddings.**
>
> Thank you very much for this suggestion. In the original manuscript **(Table 3)**, we used CLIP as a baseline. To further reinforce our findings, we additionally introduced the state-of-the-art SigLIP-2 for multimodal retrieval, and the results on MathVision are shown below; both Semantic Noise and Structural Noise are measured before Agentic Retrieval and Structural Alignment to enable a fair comparison of raw retrieval outputs. Our experiments show that CLIP and SigLIP-2 generally introduce more noise than strong unimodal encoders. This is likely because existing joint multimodal models remain weaker in parameter scale, embedding precision, or training corpus size compared with the latest unimodal encoders such as Qwen3-Embedding or the visual encoder of Qwen2.5-VL. Consequently, their embeddings may capture less fine-grained information, reducing retrieval accuracy.
> |**Text Encoder**|**Vision Encoder**|**Semantic Noise ↓**|**Structural Noise ↓**|**Effective Rate ↑**|**ICL Gains ↑**|
> |-|-|-|-|-|-|
> |**Qwen3-Embedding-4B**|**Qwen2.5-VL-VisEnc**|**0.171**|**0.584**|**0.762**|**+11.8%**|
> ||SigLIP-2|0.201|0.608|0.730|+10.1%|
> ||CLIP|0.216|0.611|0.718|+10.1%|
> |Qwen3-Embedding-0.6B|Qwen2.5-VL-VisEnc|0.194|0.595|0.749|**+11.8%**|
> ||SigLIP-2|0.209|0.617|0.727|+10.1%|
> ||CLIP|0.227|0.623|0.721|+10.1%|
> |SigLIP-2|Qwen2.5-VL-VisEnc|0.254|0.626|0.672|+8.4%|
> ||SigLIP-2|0.215|0.614|0.725|+10.1%|
> ||CLIP|0.284|0.638|0.645|+6.7%|
> |CLIP| Qwen2.5-VL-VisEnc|0.276|0.631|0.652|+8.4%|
> ||SigLIP-2|0.292|0.647|0.632|+6.7%|
> ||CLIP|0.311|0.659|0.606|+6.7%|
>
> We also find that when both the text and vision encoders come from SigLIP-2, its performance surpasses mixed configurations that combine SigLIP-2 with unimodal encoders from other model families. This supports your insight that **joint multimodal embeddings can yield more internally consistent retrieval. However, their performance still trails the strongest unimodal encoders.** Therefore, our standard implementation **adopts separate unimodal embedding models for optimal performance**. In addition, joint multimodal models like SigLIP-2 and CLIP remain practical (especially for lightweight or low-resource scenarios) and are included in our embedding model zoo. We expect ContextNav to benefit naturally from future advances in stronger joint multimodal embedding models. We have added the relevant information in Table 3 of the manuscript.
>
> >**Weakness 4 - Ablations on orchestration dynamics**
>
> Thank you very much for this comment. To further examine the orchestration dynamics, we fixed the set of tools and predefined six representative toolchain paths, from which the agent was required to choose during contextualization. We evaluated these fixed toolchains on a simple benchmark (CLEVR) and a more complex one (MathVision). Both the downstream MLLM and the policy are Gemini-2.0-flash. As shown in the table below, Accuracy refers to the downstream ICL performance obtained by applying a given toolchain across the dataset, whereas the dynamic pick rate refers to the probability that this toolchain is ultimately selected under the dynamic optimization strategy. Moreover, TR, VR, AR, and SA denote textual retrieval, visual retrieval, agentic retrieval, and structural alignment, respectively.
>
> |**Toolchain**|**CLEVR Accuracy**|**CLEVR Dynamic Pick Rate**|**MathVision Accuracy**|**MathVision Dynamic Pick Rate**|
> |-|-|-|-|-|
> |(database tools) → TR → VR → AR → SA|0.810|4.5%|0.517|15.8%|
> |(database tools) → VR → TR → AR → SA|0.790|2.0%|0.533|49.2%|
> |(database tools) → TR → AR → SA|0.775|0%|0.517|29.2%|
> |(database tools) → VR → AR → SA|0.800|15.5%|0.467|0%|
> |(database tools) → TR → VR → AR|0.810|76.0%|0.500|2.5%|
> |(database tools) → VR → TR → AR|0.790|2.0%|0.508|3.3%|
> |**Dynamic Orchestration**|**0.830**|-|**0.550**|-|
>
> Our results show **clear differences across datasets**. CLEVR prefers toolchains with late visual retrieval and no structural alignment, whereas MathVision favors late textual retrieval. Moreover, CLEVR (easy and clean) exhibits a highly concentrated pick rate on a single toolchain, while MathVision (complex and noisy) distributes its choices across multiple toolchains. These findings highlight that: (1) the optimal orchestration strategy is dataset- and query- dependent, and (2) complex or noisy datasets require more diverse toolchain selections rather than relying on a single path. **This further demonstrates the importance of dynamic optimization, which allows ContextNav to adapt across varying query types and data conditions**. We have added the relevant information in Appendix D.3 and Table 14 in the manuscript.

---

> ### Author Response · Authors · 2025-11-21
>
> # Response to Reviewer ppfZ (Part 3)
>
>
> >**Weakness 5 + Question 6 + Suggestion 4 - Statistical Analysis**
>
> Thank you very much for your suggestion. We have added error bars for Table 1 as well as Figures 3 and 4. Since directly overlaying error bars on Figures 3 and 4 slightly affects visual clarity, we present the corresponding error-bar values in tabular form (reported in Appendix B.2 and Tables 5, 6 and 7 of the revised manuscript). As these tables are **quite large and not well-suited for display on the OpenReview**, we have integrated them into the manuscript. Please refer to the updated version for the detailed modifications. We sincerely appreciate your suggestion, which has helped us make our experimental results more complete and rigorous.
>
> >**Question 1 - The diversity of the datasets**
>
> Thank you very much for this comment. In selecting our datasets, we aimed to **ensure broad task diversity and strong representativeness**. For composite-task datasets, we included:
>
> **MME-RealWorld:** challenging **comprehensive real-world** multimodal understanding (such as complex scene and object recognition, ultra-high-resolution remote sensing, charts/tables in natural scenes, traffic perception, high-resolution surveillance).
> **BlindTest:** **basic visual** tasks that are trivial for humans but difficult for MLLMs (e.g., counting intersections, circle intersection, overlapping shapes, nested squares, grid estimation).
> **CharXiv:** natural **scientific** charts from real papers, capturing scientific multimodal QA in realistic settings.
> **GVL:** **topological** multimodal reasoning tasks(connectivity, cycles, topological sort, shortest path, max flow, bipartite matching, Hamiltonian path).
> **MathVision:** **comprehensive multimodal mathematical reasoning** tasks (algebra, geometry, combinatorics, statistics, topology, etc.).
>
> For single-task datasets, we **follow the representative prior work VL-ICL**, including **CLEVR** (attribute reasoning), **FOMI** (visual concept substitution), and **TextOCR** (optical character recognition).
>
> This selection shows that, beyond the traditional single-task setups used in prior work, we extend evaluation to a much broader range of multimodal reasoning tasks. Such a comprehensive design reflects our goal of making ContextNav a generalizable framework capable of handling diverse task distributions. Detailed dataset descriptions are provided in the Appendix E.1 of our manuscript for more clarity.
>
> >**Question 2 - The noisiness of the datasets we have adopt**
>
> Thank you very much for this comment. We have provided a detailed explanation and empirical analysis in our response to **Weakness 2**, which demonstrates that the datasets and corpora we adopt are indeed highly noisy. In addition, we have included experiments on a larger-scale noisy dataset, further validating the noise robustness of ContextNav. The corresponding content has also been incorporated into the manuscript (Tables 10 and 12).
>
> >**Question 3 - Experiments on joint multimodal embeddings**
>
> Thank you very much for this comment. In the original manuscript, **Table 3** included CLIP as the joint multimodal embedding baseline, compared against unimodal embedding pairs. The results show that CLIP achieves a slightly weaker performance (+6.7% gains) compared to the stronger SOTA unimodal embedding models (+11.8% gains). In the revised manuscript, we additionally introduced SigLIP-2, a state-of-the-art joint multimodal embedding model, to further extend this experiment, which also shows slightly lower gains (+10.1%). We provide the expanded results and discuss this phenomenon in detail in our response to **Weakness 3**. We have added the relevant information in Table 3 of the manuscript.

---

> ### Author Response · Authors · 2025-11-21
>
> # Response to Reviewer ppfZ (Part 4)
>
>
> >**Question 4 - The choices of policy backbones**
>
> Thank you very much for raising this concern. In the original manuscript, we report the performance of using the Qwen2.5-VL series as the policy model in **Table 2**. In addition, we further presented related results in our response to **Weakness 1**. We include the relevant results in the table below and additionally provide the results for GPT-4o.
>
> |**Policy**|**Semantic Noise↓**|**Structural Noise↓**|**ICL Gains↑**|**System Latency (s)↓**|**Token Costs↓**|**API Pricing↓**
> |-|-|-|-|-|-|-|
> |Qwen2.5-VL-3B|0.080|0.139|+8.4%|5.10|22.2K|-|
> |Qwen2.5-VL-7B|0.073|0.107|+10.1%|7.38|22.6K|-|
> |Gemini-2.0-flash|0.053|0.084|**+11.8%**|3.26|22.5K|0.003$
> |GPT-4o|0.053|0.081|**+11.8%**|3.92|23.1K|0.08$
>
> Our decision to adopt Gemini-2.0-flash is motivated by a **balanced consideration of performance, latency, and cost**. Empirically, Gemini-2.0-flash delivers **strong performance** while achieving substantially **lower latency** than alternative models. As an API-based solution, it also avoids the computational overhead associated with local inference, with a cost of $0.003 per invocation, which is **only 1/27 of GPT-4o**, while achieving comparable downstream gains. At the same time, open-weight multimodal models in the **7B/3B range also exhibit competitive capability**. Although a small performance gap remains, these models are still efficient and practical options.
>
> Overall, these findings make Gemini-2.0-flash a natural choice as our standard policy model, offering the best balance between effectiveness and practical deployment, while it remains fully compatible with open-weight alternatives. This further demonstrates that **ContextNav does not depend on any specific proprietary model and is robust and broadly applicable with respect to policy selection.** (A secondary practical consideration is that the Gemini API provides 1,500 free calls per day, which helps further reduce the operational cost of ContextNav in real-world deployments.) We have added the relevant information in Appendix D.6 and Table 18 of the manuscript.
>
>
> >**Question 5 - The construction of OGG and the ContextNav's robustness to it**
>
> Thank you for this comment. The construction and edge-definition process of the OGG is automated. We instruct GPT-4o to infer tool composability by understanding and matching the input–output data formats of all atomic tools. Since each tool function has clearly defined I/O specifications, the automatically constructed OGG is **highly reliable** (as shown in **Table 2** of the manuscript, the toolchain execution success rate is close to **100%**). Regarding ContextNav’s robustness to the OGG design, we provide analyses from two complementary perspectives: database management and contextualization (downstream ICL gain).
>
> **Robustness of database management to OGG design.** Because database management involves transforming unstructured corpus data into structured representations, the data-flow dependencies are strict. Missing, redundant, or incorrectly ordered tools can directly result in invalid data flows. To address this, we implement engineering-level validation: if a generated toolchain triggers a data-flow exception due to unsupported or ill-formed tool sequences, we automatically regenerate the chain. Only when repeated regeneration fails do we check whether the issue arises from the structure of the OGG; if so, we apply minimal manual adjustments to correct it.
>
> **Robustness of downstream gains to OGG design.** The downstream ICL improvements primarily depend on the contexts produced during contextualization. To ensure the frameworks' robustness with respect to OGG design, we standardize the data-flow interface for contextualization tools, allowing them to be freely composed and enabling the agent to optimize their usage through memory and feedback. In practice, these tools are fully connected within the OGG (as shown in the **appendix E**), ensuring that contextualization never encounters missing or redundant tool dependencies, and thus does not negatively impact downstream ICL gains.
>
> Thank you again for your valuable comment. We have added the relevant information in Appendix E.2 of the manuscript.
>
>
> >**Question 6 - Statistical Analysis.**
>
> Thank you for this comment. In our response to **Weakness 5**, we have provided a detailed explanation of the additional error-bar analyses we incorporated to strengthen the statistical rigor of our results. We sincerely appreciate your suggestion, which has been highly valuable in improving the clarity and interpretability of our experimental presentation. We have added the relevant information in Appendix B.2 and Tables 5, 6 and 7  of the manuscript.

---

> ### Author Response · Authors · 2025-11-21
>
> # Response to Reviewer ppfZ (Part 5)
>
> >**Question 7 - Shared patterns across failure modes**
>
> Thank you for raising this concern. In our standard implementation of ContextNav, we **have not observed shared failure modes so far**. The workflow of ContextNav is **highly robust, even for ambiguous queries or highly compositional tasks** (such as MathVision and CharXiv), and it consistently achieves strong gains as shown in the Table 1 in the manuscript. This robustness stems from the strong embedding models within our standard Implementation (Qwen3-Embedding and Qwen2.5-VL-VisEnc), which produce highly accurate semantic representations. These embeddings enable us to reliably retrieve well-aligned candidate examples, even under ambiguity or complex compositional conditions. Subsequently, the MLLM policy further refines these candidates by leveraging its internal reasoning capabilities, effectively addressing remaining ambiguity or compositional complexity. This ensures that **the contextual examples delivered to the downstream MLLM are highly relevant**, making ContextNav-induced failures rare.
>
> However, we also observe a few **special failure cases**. Specifically, we observed that using a smaller model  (e.g., Qwen2.5-VL-3B) as the policy is more likely to cause it to **repeatedly generate certain tokens** in its output, until the output token budget is exhausted, and our downstream regex-based parser fails to capture a valid toolchain. This is a token-level repetition loop that smaller policies are more likely to exhibit in our observation. Based on our experience, it also tends to occur more frequently when the conditioning context is long but the required output is relatively short. As a result, the toolchain (or other tokens) could be repeatedly produced, and the EOS token may never appear. **This phenomenon is rare, it reflects a general model-level behavior rather than a system error with our framework, and therefore does not compromise the system’s robustness. When this occurs, simply prompting the policy to regenerate the toolchain could effectively resolve the issue.** As shown in the policy ablation results of Table 2 in the manuscript, the success rate across multiple attempts will be close to 100%. We have included the relevant discussion in the Appendix E.3.
>
>
> >**Suggestion 5 - ICL gain vs. token cost / latency per model**
> >(We have addressed Suggestions 1–4 in our earlier responses)
>
> Thank you for the suggestion. Following your advice, we applied the standard ContextNav implementation to Phi-3.5V, Qwen2.5-VL, Gemini-2.0-flash, and GPT-4o on BlindTest, and compared their ICL gains against token cost and latency (measured on MathVision). For latency, we report the system-level overhead introduced by ContextNav (averaged over multiple optimization iterations), excluding the downstream MLLM’s own inference cost. The results show that ContextNav incurs broadly similar overhead across all models, while stronger downstream MLLMs exhibit slightly lower latency and token cost. We attribute this to their more accurate feedback, which facilitates smoother self-optimization within ContextNav. The results show that ContextNav could complete reasoning, optimization and actions within a **short time** (less than 5s) using only a **small number of tokens** (around 30K), while achieving **strong** downstream gains, which further highlights its **efficiency and practical deployment feasibility**. We have incorporated the relevant results and discussion into the Appendix D.7 and Table 19 of the manuscript.
>
> | Downstream MLLM     | ICL Gains ↑ | System Latency (s) ↓ | Token Costs ↓ |
> |---------------------|-------------|------------------------|----------------|
> | Phi-3.5V            | 10.2%       | 5.45                   | 45.9K         |
> | Qwen2.5-VL-7B       | 14.0%       | 5.10                   | 41.4K         |
> | Gemini-2.0-flash    | 12.2%       | 4.83                   | 35.1K         |
> | GPT-4o              | 10.3%       | 4.68                   | 33.3K         |

---

### Author Response · Authors · 2025-12-02
**Summary of Rebuttal**

>**Summary of Score Changes During the Discussion Phase**

We are deeply grateful to our Area Chairs for the substantial time and effort dedicated to our submission under such unusual circumstance. For your reference, we provide a brief summary of the score changes **(the overall score was raised to "6" on 22 Nov)** prior to the circumstance:

1. Reviewer ppfZ keeps the **score of 8**.

2. Reviewer zbjE increases the **score to 8**, with the confidence increased to 4 (**updated on 22 Nov**).

3. Reviewer q6J8 temporarily keeps the **score of 4**, while **explicitly emphasising that the reviewer would not object** if the paper is accepted.

4. Reviewer tPU8 temporarily keeps the **score of 4**, while **also explicitly emphasising that the reviewer would not object** if the paper is accepted. The reviewer also mentioned in the initial review that if the concerns are addressed, the reviewer **would definitely consider increasing the score**.

We provide the screenshots of the score updates as supporting evidence through this [[anonymous link]](https://anonymous.4open.science/r/3215a-4DC8/).

>**Summary of the Reviews and Rebuttal Progress**

We sincerely appreciate the reviewers’ recognition of our work in terms of **novelty and originality** (all reviewers), **comprehensive experiments** (all reviewers), **technical soundness** (all reviewers), **presentation quality** (Reviewers ppfZ, q6J8, tPU8), and **SOTA performance** (Reviewers ppfZ, q6J8, tPU8).

We are also grateful that the reviewers highlight our contribution in being the first to propose an agentic framework for multimodal ICL as **“a substantive contribution to the evolution of multimodal ICL, providing both scientific insight and practical implications for ICL advancement”** (Reviewer ppfZ), **“interesting, timely, and reasonable”** (Reviewer zbjE), **“exploring the largely unexplored role of agent-based methods in multimodal ICL with consistent and notable improvements over baselines”** (Reviewer q6J8), and **“focused on an underexplored direction and presenting a neat twist”** (Reviewer tPU8).

During the rebuttal and discussion phases, we receive **a total of 40 comments** (including initial weaknesses, questions, suggestions, and discussion-phase follow-ups). Among these, **38 have been fully addressed to the reviewers’ satisfaction, or without any additional concerns raised, and the rivised paper has been uploaded**. For the remaining 2 points (one from Reviewer q6J8 and one from Reviewer tPU8), we have also provided detailed responses, but since the reviewers are currently unable to comment, no further discussion has yet been made.

We are also grateful that both Reviewers q6J8 and tPU8 **explicitly and positively state in their comments that they would not object if the paper is accepted**, and that Reviewer tPU8 further explicitly indicate the reviewer **"would definitely consider increasing the score"** after the concerns are addressed.

>**Rebuttal Details**

We summarise below the specific responses and revisions we have provided to address each reviewer’s concerns. **All corresponding content have been incorporated into the revised manuscript and uploaded.**

**For Reviewer ppfZ**, we conduct empirical analyses on smaller and open-weight policies, dataset noisiness, and robustness on large-scale real-world corpora. We also further evaluate joint multimodal embeddings, analysed ICL gain–cost/latency trade-offs, and add statistical analysis together with discussions on dataset diversity and shared failure patterns. Reviewer ppfZ dose not raise any further concerns.

**For Reviewer zbjE**, we expand the comparative experiments, improve the presentation, and provide empirical analyses on toolchain optimization, semantic equivalence in structural alignment, and the outcome of extended random sampling. We also discuss dataset-splitting practices and present prompts for evaluating semantic and structural noise. Reviewer zbjE is satisfied with all our responses.

**For Reviewer q6J8**, we provide empirical and theoretical analyses on the nature of multimodal ICL, covering example selection, number, and ordering, as well as directions for RL-based extensions. The reviewer acknowledges our experiments and clarifications and encourages further analysis. Following this, we additionally examine modality dominance and validate it from an agent-based optimization perspective. Due to system limitation, we have not yet received a further response.

**For Reviewer tPU8**, we conduct empirical studies on CoT and Memory, robustness under smaller policies and hallucination, real-time performance, extended Embodied QA tasks, and inference latency/costs, and also develop a Colab-friendly code demo as suggested. The reviewer acknowledges our responses and suggests deeper clarification on the relationship between Memory and optimization process, for which we provided a detailed follow-up. Due to system limitation, we have not yet received a further response.

---

### Meta-Review · Area_Chair_Yuxs · 2026-01-06

**Summary:**

The paper introduced an agentic pipeline that automatically retrieves and verifies relevant examples from a pre-built database for in-context learning, enabling multimodal LLM to quickly adapt to novel tasks. The reviewers unanimously found the topic timely and appreciate the novelty of the tasks. The major concerns from the reviewers were (i) the lack of analysis on why the framework works, (ii) not enough empirical evidences from different models, and (iii) missing technical details (eg, computational costs). During the discussion period, the authors provided extensive experiments on new models and new experimental setups, addressing most concerns. At the end, one reviewer raised their score from 4 to 8, while the rest remains the same (but stressing that they will not be upset if the paper is accepted). While the AC agrees with the reviewers that the paper is more of an engineering effort and could be improved further if provided with more insights, the AC also recognizes that in todays era building a working pipeline is equally important and the empirical findings has its merits. The AC thus decide to accept the paper. The AC urges the authors to incorporate the feedbacks from the reviewers into their final version.

**Reviewer Concerns:**

See above.

**Reviewer Scores:**

See above.

---

### Decision · Program_Chairs · 2026-01-26

Accept (Poster)